# Soluble PD-L1 generated by endogenous retroelement exaptation is a receptor antagonist

Kevin W Ng[1], Jan Attig[1], George R Young[2], Eleonora Ottina[1], Spyros I Papamichos[3], Ioannis Kotsianidis[3], George Kassiotis[1,4]*

[1]Retroviral Immunology, The Francis Crick Institute, London, United Kingdom; [2]Retrovirus-Host Interactions, The Francis Crick Institute, London, United Kingdom; [3]Department of Haematology, Democritus University of Thrace Medical School, Alexandroupolis, Greece; [4]Department of Medicine, Faculty of Medicine, Imperial College London, London, United Kingdom

**Abstract** Immune regulation is a finely balanced process of positive and negative signals. PD-L1 and its receptor PD-1 are critical regulators of autoimmune, antiviral and antitumoural T cell responses. Although the function of its predominant membrane-bound form is well established, the source and biological activity of soluble PD-L1 (sPD-L1) remain incompletely understood. Here, we show that sPD-L1 in human healthy tissues and tumours is produced by exaptation of an intronic *LINE-2A* (*L2A*) endogenous retroelement in the *CD274* gene, encoding PD-L1, which causes omission of the transmembrane domain and the regulatory sequence in the canonical 3' untranslated region. The alternatively spliced *CD274-L2A* transcript forms the major source of sPD-L1 and is highly conserved in hominids, but lost in mice and a few related species. Importantly, *CD274-L2A*-encoded sPD-L1 lacks measurable T cell inhibitory activity. Instead, it functions as a receptor antagonist, blocking the inhibitory activity of PD-L1 bound on cellular or exosomal membranes.

*For correspondence:
george.kassiotis@crick.ac.uk

**Competing interests:** The authors declare that no competing interests exist.

## Introduction

First identified as a marker of developing thymocytes, co-inhibitory receptor programmed cell death protein 1 (PD-1) plays a well-recognised role in restraining T cell responses in autoimmunity, infection or cancer (*Chamoto et al., 2017*; *Dai et al., 2014*; *Sharpe and Pauken, 2018*; *Sun et al., 2018*). PD-1 is induced in T cells proportionally to the strength of T cell receptor (TCR) stimulation, rendering them receptive to signals from its two ligands, PD-1 ligand 1 (PD-L1) and 2 (PD-L2), both of which are membrane-bound proteins (*Chamoto et al., 2017*; *Sharpe and Pauken, 2018*; *Sun et al., 2018*).

In addition to its major membrane-bound form, a soluble form of PD-L1 (sPD-L1) has long been documented in healthy human serum and found elevated in autoimmune disease and in cancer (*Chen et al., 2011*; *Frigola et al., 2011*; *Koukourakis et al., 2018*; *Okuma et al., 2017*; *Rossille et al., 2014*; *Wan et al., 2006*; *Wang et al., 2015*; *Zhou et al., 2017*; *Zhu and Lang, 2017*). However, the precise source of sPD-L1 has remained uncertain. Cell-free PD-L1 is also found in exosomes, where it is still membrane-bound (*Chen et al., 2018*; *Poggio et al., 2019*; *Ricklefs et al., 2018*) and this exosomal PD-L1 (exPD-L1) has confounded detection of membrane-free sPD-L1. Nevertheless, membrane-free non-exosomal sPD-L1 has been demonstrated as a distinct, lower molecular weight, form of PD-L1 (*Chen et al., 2011*; *Frigola et al., 2011*).

Proteolytic cleavage and release of the extracellular domain of PD-L1 has been considered as one possible source of sPD-L1, implicating matrix metalloproteinase activity (*Chen et al., 2011*).

Membrane-free forms of sPD-L1 may also be produced by alternative splicing of the *CD274* transcript (encoding PD-L1). At least two distinct types of splicing events have been described in several recent reports to remove or affect the exon encoding the PD-L1 transmembrane domain. The first involves mid-exon splicing (*Gong et al., 2019*; *Zhou et al., 2017*), whereas the second is created by alternative polyadenylation (*Hassounah et al., 2019*; *Mahoney et al., 2019*; *Singh et al., 2018*). However, the balance between the various *CD274* isoforms and, consequently, their relative contribution to the pool of sPD-L1 remain unknown.

Also unclear is the biological activity of sPD-L1 (*Zhu and Lang, 2017*). Serum levels of sPD-L1 have been negatively associated with overall survival or response to immunotherapy in diverse cancer types, including renal cell carcinoma, diffuse large B-cell lymphoma, multiple myeloma, melanoma, and lung cancer (*Frigola et al., 2012*; *Frigola et al., 2011*; *Koukourakis et al., 2018*; *Okuma et al., 2017*; *Rossille et al., 2014*; *Wang et al., 2015*; *Zhou et al., 2017*), suggesting a possible inhibitory effect. However, immune suppression mediated by cell-free PD-L1, as well as its negative association with overall survival and response to anti-PD-1 immunotherapy has recently been attributed to exPD-L1 in melanoma, glioblastoma, and mouse models (*Chen et al., 2018*; *Poggio et al., 2019*; *Ricklefs et al., 2018*). In contrast, a study of melanoma patients did not support an inhibitory role for membrane-free sPD-L1 (*Chen et al., 2018*). Several studies have reported that, in direct in vitro assays, sPD-L1 suppresses T cell activation (*Frigola et al., 2011*; *Hassounah et al., 2019*; *Mahoney et al., 2019*; *Zhou et al., 2017*), suggesting it retains the inhibitory activity of the membrane-bound form. However, sPD-L1 completely lacked inhibitory activity in similar in vitro assays in other reports (*Chen et al., 2018*; *Gong et al., 2019*). Thus, despite its potential importance, the biological activity of sPD-L1 has not yet been established.

We have been studying the contribution of endogenous retroelements (EREs) to the diversification of the human transcriptome (*Attig et al., 2019*). Abundant genomic integrations of EREs, including long and short interspersed nuclear elements (LINEs and SINEs, respectively) and endogenous retroviruses (ERVs) (*Lander et al., 2001*) can generate alternative transcript isoforms through the supply of alternative promoters, splicing, or polyadenylation sites (*Babaian and Mager, 2016*; *Burns and Boeke, 2012*; *Feschotte and Gilbert, 2012*; *Kassiotis and Stoye, 2016*). Here, we describe *CD274* isoforms generated by transcriptional inclusion of EREs. We show that exonisation of an intronic germline LINE integration in the *CD274* gene is responsible for alternative polyadenylation of a truncated *CD274* mRNA and for production of sPD-L1. We provide further evidence that sPD-L1, produced by LINE exaptation, is evolutionarily conserved in humans, lacks inhibitory activity and is, in fact, a receptor antagonist.

## Results

### *CD274* splice variants generated by retroelement exonisation

In an effort to identify aberrant inclusion of EREs in transcripts of cellular genes, we de novo assembled transcripts expressed in a multitude of human cancers, where ERE transcriptional activity is elevated (*Attig et al., 2019*). Together with numerous retroelements (*Figure 1A*), the *CD274* locus comprises four currently RefSeq or GENCODE annotated variants (*Figure 1B*; variants 1–4) and two recently cloned variants (*Zhou et al., 2017*), each generated by one of the two variant 3 splicing alternatives (*Figure 1C*; variants 9 and 12). Inspection of our recent assembly (*Attig et al., 2019*) for ERE-overlapping transcripts at this locus identified three variants that use a terminal ERE instead of the canonical termination and polyadenylation site, referred to here as *CD274-MIRB*, *CD274-FLAM_A* and *CD274-L2A*, according to the superfamily of their respective ERE (*Figure 1D*). Transcript *CD274-MIRB* omits the splice site at the end of the canonical exon 6 and terminates at a *MIRB* SINE integrated in intron 6 (*Figure 1D*). Transcript *CD274-FLAM_A* skips the canonical terminal exon 7 and instead uses an alternative splice acceptor site at a downstream *FLAM_A* SINE (*Figure 1D*). Transcript *CD274-L2A* corresponds to *CD274* splice variant 4 (NCBI accession NM_001314029).

In contrast to transcripts *CD274-MIRB* and *CD274-FLAM_A*, which include the transmembrane domain encoded by the canonical exon 5, transcript *CD274-L2A* encodes a truncated isoform of PD-L1, which retains the extracellular Ig-like V and C2 type domains that mediate receptor binding (*Figure 1D,E*; *Figure 1—figure supplement 1A–D*) and has recently been reported to produce sPD-

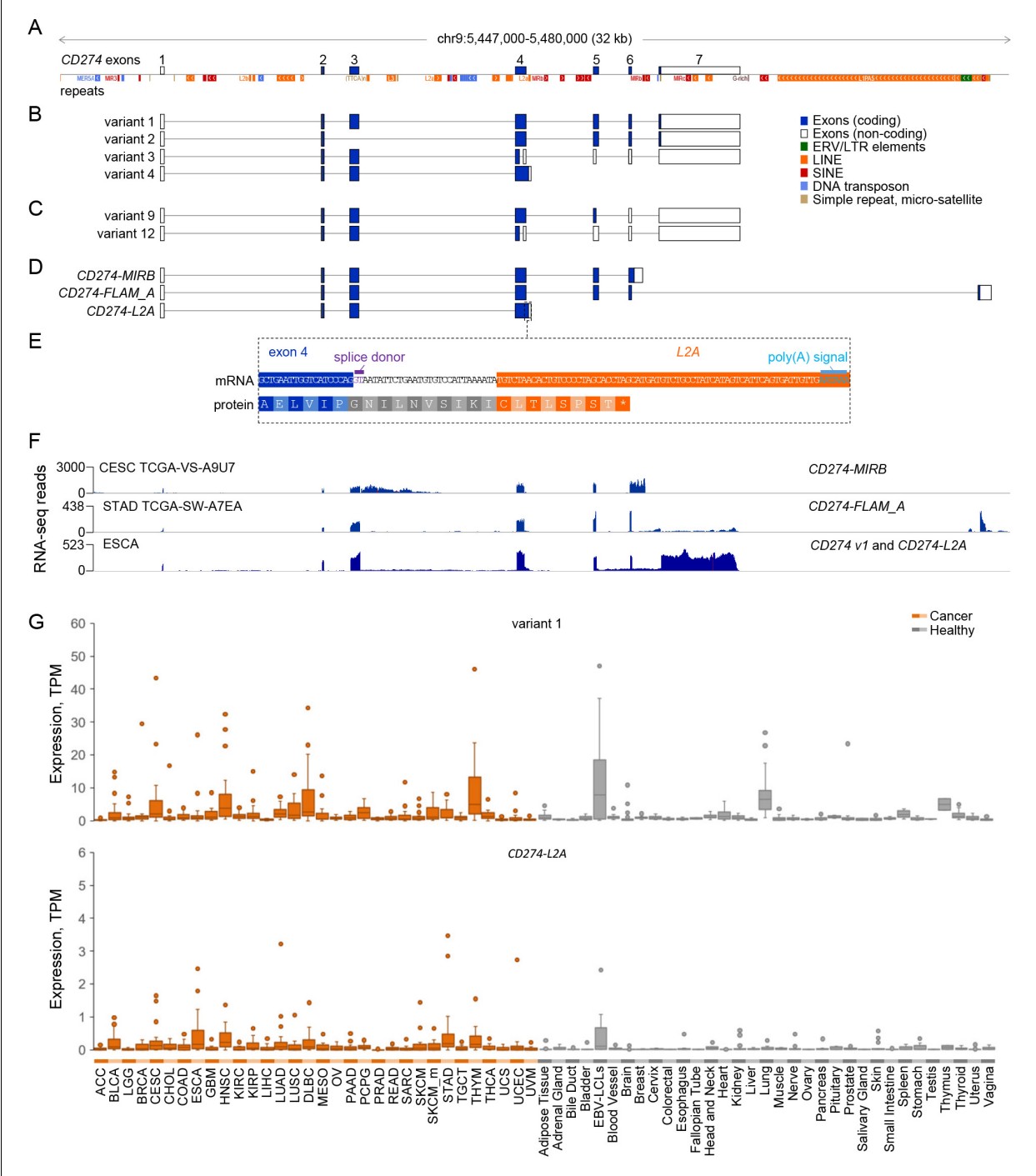

**Figure 1.** CD274 protein domains and splice variants. (**A**) Depiction of reference genome EREs and other repeats in the genomic locus spanning the *CD274* gene, relative to *CD274* exons. (**B**) GENCODE or RefSeq annotated splice variants of the *CD274* gene. Variant numbers correspond to the following NCBI Reference Sequence accessions: variant 1 (NM_014143), variant 2 (NM_001267706), variant 3 (NR_052005) and variant 4 (NM_001314029). (**C**) Recently-described novel variants encoding sPD-L1 (*Zhou et al., 2017*). (**D**) *CD274* splice variants de novo assembled in this study and overlapping one or more EREs. (**E**) Inclusion of an *L2A* element as a terminal exon and polyadenylation site in splice variant *CD274-L2A*. The relative position of the intronic *L2A* element, as well as the novel C-terminal amino acid created from its exonisation are also indicated. (**F**) RNA-seq traces representative of each of the ERE-overlapping *CD274* variants, *CD274-MIRB*, *CD274-FLAM_A* and *CD274-L2A*. For the low recurrence *CD274-MIRB* and *CD274-FLAM_A* variants, the samples with the highest expression are shown, whereas for the high recurrence *CD274-L2A*, a representative ESCA sample is shown. (**G**) Box plot of *CD274* variant 1 and *CD274-L2A* expression (in TPMs) in the indicated cancer patient (n = 24 for each indication) and healthy control samples (n between 2 and 156).

*Figure 1 continued on next page*

*Figure 1 continued*

The online version of this article includes the following source data and figure supplement(s) for figure 1:

**Source data 1.** Expression of CD274 variant 1 and CD274-L2A in TCGA and GTEx samples.
**Figure supplement 1.** The *CD274-L2A* protein product retains receptor binding domains but not transmembrane domain.
**Figure supplement 2.** The *L2A* element in intron 4 of the *CD274* gene is essential for sPD-L1 production.
**Figure supplement 3.** Expression of *CD274-MIRB* and *CD274-FLAM_A* variants.
**Figure supplement 3—source data 1.** Expression of CD274-MIRB and CD274-FLAM_A in TCGA and GTEx samples.
**Figure supplement 4.** Splice junction analysis of *CD274* variant expression.

L1 with the capacity to bind PD-1 (*Hassounah et al., 2019*; *Mahoney et al., 2019*). We found that the *CD274-L2A* variant omits the splice site at the end of the canonical exon 4 and instead continues into an *L2A* LINE integration in intron 4, which acts as a terminal exon and polyadenylation signal (*Figure 1D,E*). Thus, this form of sPD-L1 is generated by alternative splicing and *L2A* element exonisation following the canonical exon 4, resulting in a novel 18-amino acid C-terminal sequence (*Figure 1D,E*). Production of sPD-L1 by this transcript critically depended on the presence of the *L2A* element in intron 4, as its deletion abolished sPD-L1 expression by an intron 4-containing minigene (*Figure 1—figure supplement 2A–C*).

Expression of *CD274* variant 1, encoding the canonical full-length PD-L1, was detected in a variety of healthy tissues and was further upregulated in multiple cancers (*Figure 1F*), as expected (*Sharpe and Pauken, 2018*; *Sun et al., 2018*). In contrast, *CD274-MIRB* and *CD274-FLAM_A* were expressed at high levels only in a few patient samples (*Figure 1—figure supplement 3*). Although no structural variations that could account for the *CD274-MIRB* and *CD274-FLAM_A* transcript structures were found in the highest expressing samples, it is likely that these transcripts arise from processes specific to the individual tumours and indicate the sensitivity of our assembly in capturing transcripts expressed in only a few individuals.

Consistent with two recent reports (*Hassounah et al., 2019*; *Mahoney et al., 2019*), variant *CD274-L2A* was readily detected in a variety of healthy tissues and cancer samples (*Figure 1G*). Similarly to variant 1, *CD274-L2A* was upregulated in multiple cancers, although levels of *CD274-L2A* expression levels remained overall 10 times lower than of variant 1 (*Figure 1G*). In addition to *CD274-L2A*, variants 3, 9, and 12 also encode truncated forms of PD-L1 lacking the transmembrane domain, due to shared mid-exon splicing events (*Gong et al., 2019*; *Zhou et al., 2017*) (*Figure 1—figure supplement 1B,C*). We therefore examined the relative contribution of all these distinct transcripts to potential sPD-L1 production by comparing their expression levels. Transcripts corresponding to variants 9 and 12 were neither present in our assembly nor were they supported by manual splice junction analysis (*Figure 1—figure supplement 4*). These observations suggest that, similarly to *CD274-MIRB* and *CD274-FLAM_A*, variants 9 and 12 were sporadically expressed in our sample collection or in independent cohorts (*Gong et al., 2019*) and that *CD274-L2A* is the predominant sPD-L1-encoding variant in the majority of tumour and healthy samples.

## CD274-L2A genomic features are highly conserved in hominids

We reasoned that if the *L2A* exonisation that generates the *CD274-L2A* isoform were not accidental, but produced a molecule with a distinct biological function, there might be evidence for evolutionary selection of the genomic features that permit the generation of this transcript. Since sPD-L1 has not been described in laboratory mice and was not detectable in murine cells lines that readily express membrane bound PD-L1 (*Figure 2—figure supplement 1A,B*), we first considered the possibility that generation of the *CD274-L2A* transcript is specific to humans and related species. Although no *L2A* integration is annotated by RepeatMasker (www.repeatmasker.org) in intron 4 of the murine *Cd274* gene, the major expansion of *L2A* elements in mammalian genomes is thought to have preceded placental mammal diversification and has now ceased (*Goodier and Kazazian, 2008*). It was, therefore, possible that an ancestral *L2A* integration was present in all mammals but was subsequently modified or lost, according to species-specific evolutionary paths of the *CD274* locus.

Comparative genomics revealed surprisingly good alignment of this intronic region across all mammals (*Figure 2A,B*), providing evidence for an ancestral *L2A* integration. This alignment was disrupted by a few insertions and deletions in distinct species (*Figure 2B–E*). The largest insertion was

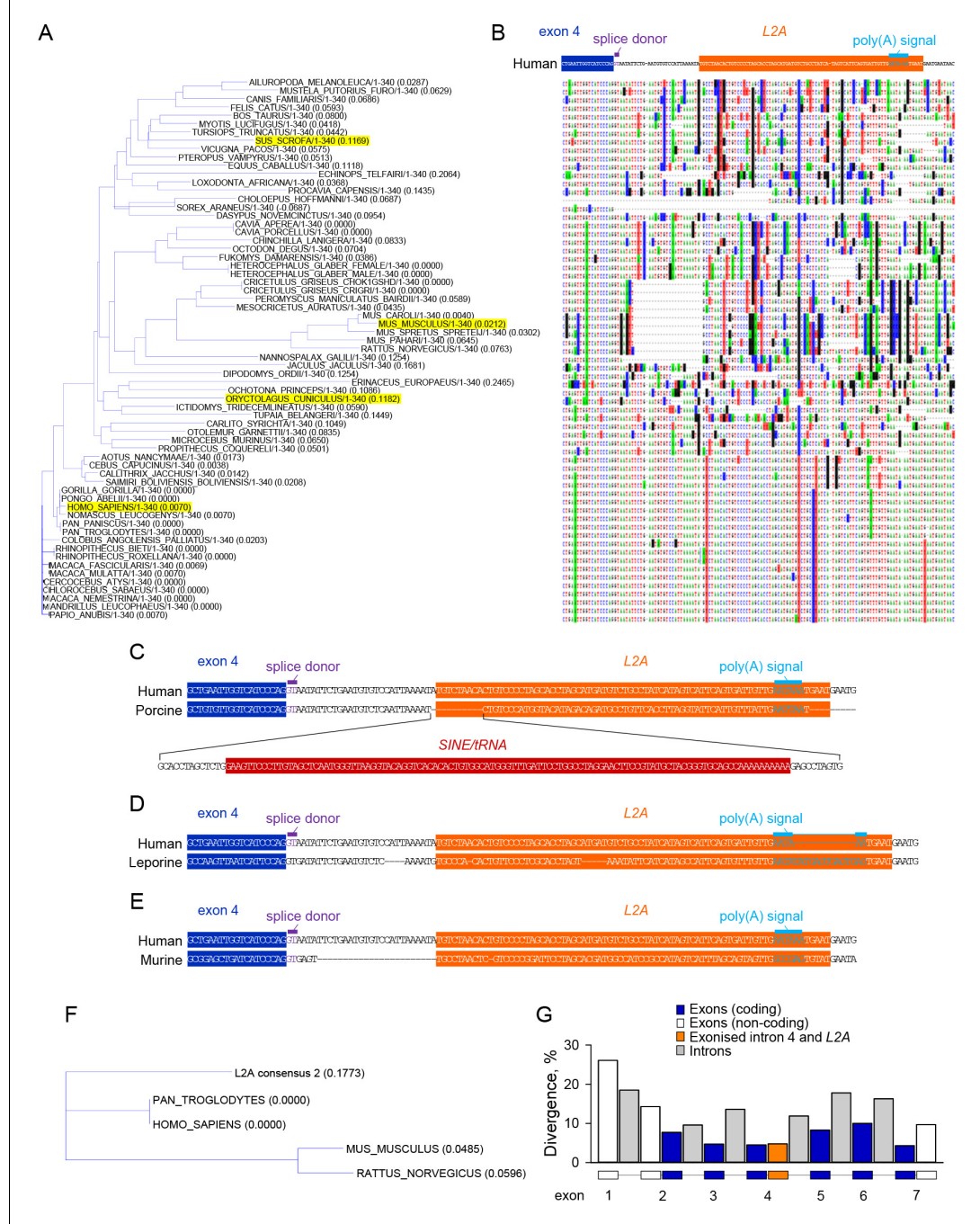

**Figure 2.** Evolutionary conservation of *CD274-L2A* genomic features in hominids. (A–B) Genomic alignment of the indicated portion of the *CD274* gene using nucleotide sequences from 65 eutherian mammals. A large SINE insertion in the porcine *Cd274* gene and a smaller insertion in the leporine *Cd274* gene were removed to aid the visual representation of alignment (both these species are highlighted in A. Base substitutions are indicated by highlighting and absence of highlighting denotes base conservation. (C) Comparison of the human and porcine genes, illustrating the SINE/tRNA insertion in the latter. (D) Comparison of the human and leporine genes, illustrating an insertion in the polyadenylation site of the latter. (E) Comparison of the human and murine genes, illustrating a 24-nucleotide deletion in the latter and mutations at the splice and polyadenylation sites. (F) Alignment tree depicting the distance of the consensus *L2A* element sequence from the respective human, chimpanzee, murine and rat elements. (G) Sequence divergence of coding and non-coding exons, introns and of the 100 nucleotides covering the exonised part of intron 4 and embedded *L2A* element in *CD274* genomic sequences from 10 primate species. The individual segments of the *CD274* gene compared were scaled to the same width. The online version of this article includes the following figure supplement(s) for figure 2:

**Figure supplement 1.** Expression of *CD274-L2A* and production of sPD-L1 in mammalian species.

caused by a secondary *SINE* integration within the *L2A* integration specific to the porcine *Cd274* gene (*Figure 2C*). This insertion did not affect the splice donor site at the end of exon 4 of the *L2A* polyadenylation signal (*Figure 2C*), but instead modified the last 9 of the novel 18-amino acid C-terminal sequence (GNILNVSIKMHLALEVPL). In the leporine *Cd274* gene, a smaller insertion disrupted the *L2A* polyadenylation signal (*Figure 2D*), which likely compromises mRNA stability.

Of note, the *Cd274* gene in mice, rats, and hamsters exhibited a 24-nucleotide deletion in this intronic region, as well as numerous substitutions (*Figure 2A,B*). These changes seemed to have two major consequences. Firstly, in contrast to the splice donor site at the end of the human *CD274* exon 4, which is predicted to be a weak motif (GUAAUA), the equivalent splice donor site in the murine *Cd274* gene (GUGAGU) was identical to the intronic part of the consensus splice donor motif GUPuAGU (*Figure 2E*). Secondly, the polyadenylation signal in the murine *L2A* integration was mutated and no longer appeared functional (*Figure 2E*). Therefore, humans, non-human primates and other species, but not rabbits or rodents, such as mice, rats and hamsters, have retained the properties required to produce the *CD274-L2A* transcript. Accordingly, *CD274-L2A* transcripts were detected by qRT-PCR in non-human primate cells, but not in cells of rabbit or mouse origin, despite expression of the full-length *CD274* in all these cell lines (*Figure 2—figure supplement 1C,D*). Together with the high degree of conservation in the hominid lineage, these observations suggest that the ability to produce the *CD274-L2A* transcript has been selected for, likely through an important biological function. The ancestral *L2A* integration was better preserved in hominids than in rodents, as suggested by greater sequence homology of the human and chimpanzee *L2A* integration (67.1%) with the consensus *L2A* than either the rat or mouse *L2A* integrations (49.4% and 45.7%, respectively) (*Figure 2F*). This difference was likely caused by faster evolution of the rodent *L2A* integration, as the human *L2A* integration in the *CD274* intron 4 appeared to evolve overall at a similar rate as the average of other *L2A* integrations of comparable size in the human genome (divergence: 30.3%; deletions: 7.3%; insertions: 0.0% for the *CD274* intron 4 *L2A* integration, and divergence: 27.3%; deletions: 7.8%; insertions: 5.0% for all 437 *L2A* integrations of 75–77 nucleotides in the human genome). However, potential exaptation of the *L2A* element in the *CD274* intron 4 would select for the new function for this *L2A* element and, consequently, only for the features that are necessary for this new function. These features included the splice site at the end of the *CD274* exon 4, the sequence encoding the novel 18 C-terminal amino acids and the polyadenylation site and were provided jointly by the first 29 nucleotides of intron 4 and the first 71 nucleotides of the succeeding *L2A* element. Indeed, when the sequence divergence of these 100 nucleotides between the splice site at the end of the *CD274* exon 4 and the polyadenylation site in the *L2A* element was examined in 10 primate species, it was found as low as that of coding exons, contrasting the higher divergence of the rest of *CD274* intron 4, other *CD274* introns and non-coding exons (*Figure 2G*). These findings indicate retention of ability to produce *CD274-L2A* in primates and certain other species, but its loss through specific mutation in rabbits and rodents.

## CD274-L2A regulated independently from the canonical variant 1

The expression pattern of the two main *CD274* variants, variant 1 and *CD274-L2A*, suggested a degree of co-regulation (*Figure 1G*), which was expected given they are both driven by the same promoter. However, shifts in splicing patterns would create one variant transcript at the expense of the other, and regulatory sequences in the 3' untranslated region (UTR) of variant 1 (*Coelho et al., 2017*), but not of *CD274-L2A*, could affect their stability differentially. Indeed, expression of the two transcripts only weakly correlated in healthy tissues ($R^2 = 0.138$) or different cancer types ($R^2 = 0.327$) (*Figure 3A*). Importantly, the ratio of *CD274-L2A* to variant 1 was generally increased in cancer, as exemplified when comparing healthy lung to lung squamous cell carcinoma (LUSC) or lung adenocarcinoma (LUAD) (*Figure 3A*). Comparable results were obtained when individual cancer of healthy tissue types were examined separately (*Figure 3—figure supplement 1*). For example, healthy lung samples expressed almost exclusively variant 1, whereas LUAD acquired expression also of *CD274-L2A* (*Figure 3—figure supplement 1*). Similarly, expression of the two transcripts in 933 cancer cell lines was weakly correlated ($R^2 = 0.215$), with their ratios varying between cell lines by at least two orders of magnitude (*Figure 3B*).

Although the expression patterns of variant 1 and *CD274-L2A* observed here were broadly concordant with those independently reported (*Hassounah et al., 2019*; *Mahoney et al., 2019*), our data did not support a strong correlation between the two transcripts. Such differences are likely

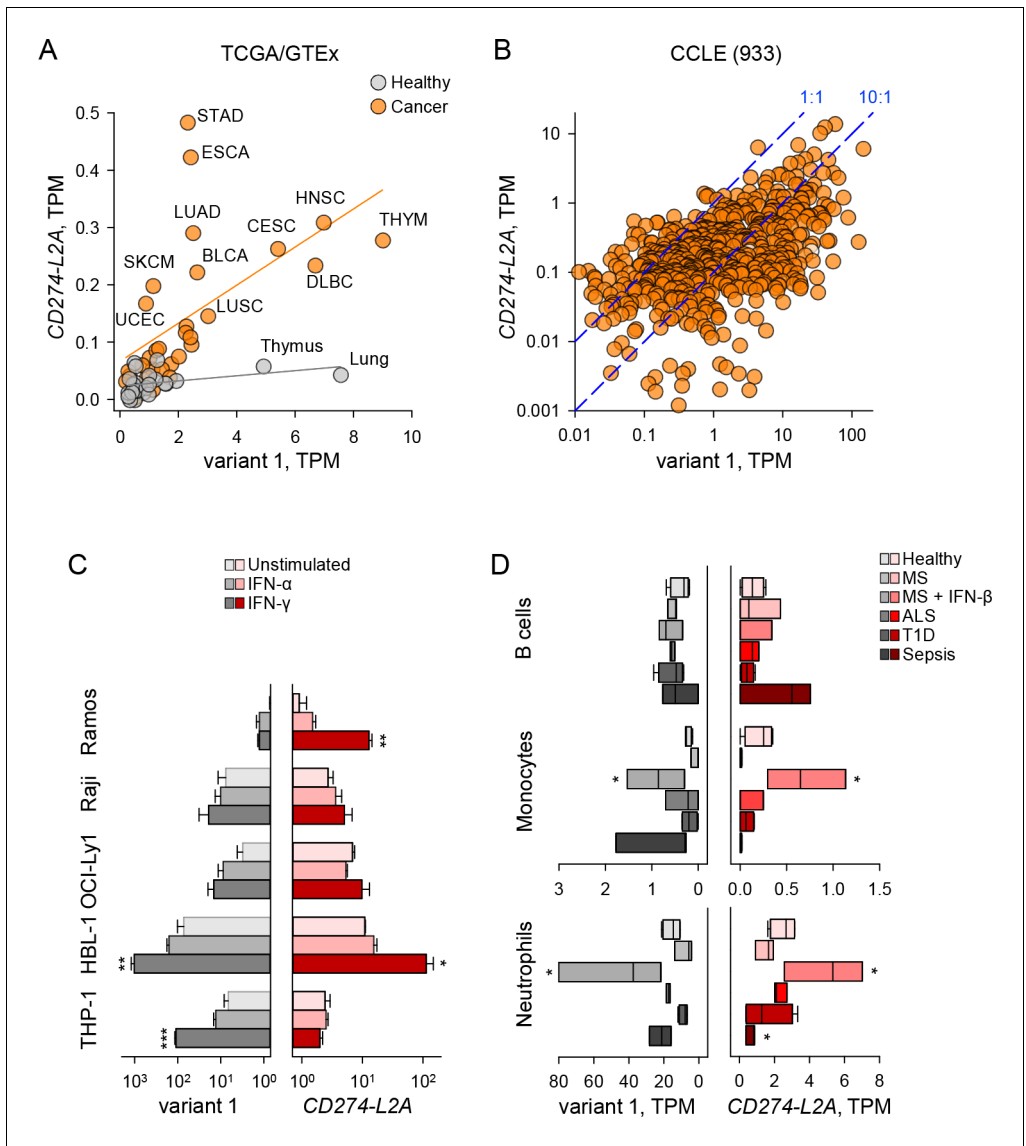

**Figure 3.** *CD274-L2A* and *CD274v1* expression are decoupled under certain stimuli. (**A**) *CD274* variant 1 and *CD274-L2A* expression across TCGA tumour and GTEx healthy samples. Average TPM is shown per tissue type, with linear regression performed separately for tumour and healthy samples. (**B**) *CD274* variant 1 and *CD274-L2A* expression (in TPMs) across the CCLE dataset. Dashed lines denote a 10:1 and 1:1 ratio of *CD274v1:CD274-L2A*, respectively. (**C**) Expression of *CD274* variant 1 and *CD274-L2A*, measured by qRT-PCR using variant-specific primers in five leukocyte cell lines. Cells were stimulated with IFN-α or IFN-γ for 48 hr or were left untreated. Mean (± SEM) expression normalized to *HPRT* from three independent experiments is shown. (**D**) Expression of *CD274* variant 1 and *CD274-L2A* (in TPMs), calculated using RNA-seq data (SRP045500) from B cells, monocytes and neutrophils isolated from peripheral blood of healthy individuals of patients with Sepsis, ALS or T1D or from MS patients before and 24 hr after the first treatment with IFN-β.

The online version of this article includes the following source data and figure supplement(s) for figure 3:

**Source data 1.** Expression of CD274 variants in the CCLE collection.
**Figure supplement 1.** *CD274-L2A* and *CD274v1* expression in individual cancer and healthy samples.
**Figure supplement 2.** *CD274-L2A*-derived soluble PD-L1 is exosome independent.

due to distinct methods used for quantitation of these two *CD274* variants in RNA-seq data in the different studies. For example, reads mapping uniquely to *CD274-L2A* or a proxy for *CD274-L2A* expression (ratio of shared exon 4 to non-shared exon 5) were previously used to calculated expression as reads per kilobase per million reads (RPKM) (*Hassounah et al., 2019*; *Mahoney et al.,*

*2019*). In contrast, we used an extended transcriptome assembly, which includes additional exon-sharing transcripts produced by the *CD274* locus and calculated expression as transcripts per million (TPM), a modification of RPKM measurements that was developed to eliminate inconsistent calculations across samples (*Wagner et al., 2012*).

We therefore examined in more detail the possible correlation of variant 1 and *CD274-L2A* expression using variant-specific qRT-PCR at the steady-state, as well as following interferon (IFN) stimulation. To this end, we used a series of B cell leukaemia cell lines, representing progressive steps of B cell differentiation and concomitant PD-L1 expression (*Basso and Dalla-Favera, 2015*). Indeed, in the absence of IFN stimulation, the Burkitt's lymphoma Ramos cells transcribed much lower amounts of either transcript than the activated B cell-like diffuse large B cell lymphoma (DLBCL) HBL-1 cells, whereas Burkitt's lymphoma Raji and germinal centre B cell-like DLBCL OCI-Ly1 cells were intermediate (*Figure 3C*). Also intermediate were monocytic THP-1 cells, which were also used for comparison (*Figure 3C*).

As with analysis of RNA-seq data, the steady-state ratios of the two forms between the various cell lines varied by at least one order of magnitude when analysed by qRT-PCR, with OCI-Ly1 expressing more *CD274-L2A* than variant 1 (*Figure 3C*). Both variants were weakly and comparably responsive to IFN-α stimulation (*Figure 3C*). Notably, however, expression of the two forms was strongly, but not always equally, responsive to IFN-γ stimulation. Indeed, IFN-γ stimulation-induced expression predominantly of variant 1 in THP-1 cells and of the *CD274-L2A* variant in Ramos cells (*Figure 3C*).

As copy number, as well as transcriptional regulation of the *CD274* gene may be altered in cancer cell lines, particularly in leukaemias (*Green et al., 2010*; *Rosenwald et al., 2003*; *Wessendorf et al., 2007*), we next investigated variant 1 and *CD274-L2A* expression in healthy immune cells. For this purpose, we used RNA-seq data (SRP045500), generated from human primary leukocyte subsets isolated from peripheral blood of healthy individuals and those with sepsis, Amyotrophic Lateral Sclerosis (ALS), Type 1 Diabetes (T1D), or Multiple Sclerosis (MS) (*Linsley et al., 2014*). Sepsis is associated with elevated serum levels of IFNs (*Schulte et al., 2013*) and MS patients are treated with recombinant IFN-β, allowing analysis of the in vivo effect of IFNs on PD-L1 variant 1 and *CD274-L2A* expression. B cells expressed moderate levels of either isoform in healthy individuals and upregulated *CD274-L2A* expression in 1epsis (*Figure 3D*). Expression of variant 1 and *CD274-L2A* followed a similar pattern in monocytes and neutrophils, although expression of both isoforms was on average 10-times higher in neutrophils than monocytes or B cells (*Figure 3D*). Both monocytes and neutrophils displayed strongly elevated expression of variant 1 and *CD274-L2A* following IFN-β treatment of MS patients, with monocytes expressing the two variants at nearly equal levels (*Figure 3D*). In contrast, expression of *CD274-L2A* was disproportionately reduced in monocytes and neutrophils isolated from sepsis patients, in comparison with the same cell types from healthy individuals, whereas expression of variant 1 remained elevated (*Figure 3D*). As a result, the ratio of variant 1 to *CD274-L2A* was elevated in monocytes and neutrophils from sepsis patients (32 and 34, respectively), in comparison with B cells from the same patients, where it was inverted (0.6) ($p \leq 0.004$, one-way ANOVA). Together, these data highlighted a certain degree of independent regulation of the two transcript variants, as opposed to a fixed rate of aberrant splicing creating the *CD274-L2A* isoform as a by-product of the canonical variant 1, particularly following IFN stimulation in vitro and in vivo.

## CD274-L2A protein product lacks suppressive activity

To investigate the possible biological function that might account for the evolutionary conservation of the *CD274-L2A* transcript features in hominids, as well as its transcriptional regulation, we first examined the suppressive activity of its protein product. To obtain a source of *CD274-L2A*-derived sPD-L1 free from exPD-L1 or other potential sPD-L1 forms, we transduced murine B-3T3 cells (which naturally lack the *CD274* gene) and HEK293T cells (which do not express their endogenous *CD274* gene) with a *CD274-L2A*-expressing retroviral vector. Both B-3T3.*CD274-L2A* and HEK293T.*CD274-L2A* cell lines produced readily detectable levels of sPD-L1 at 10–40 times higher amounts than those detected by ELISA in supernatants of IFN-γ stimulated HBL-1 cells (*Figure 4A*; *Figure 3—figure supplement 2A*). Importantly, supernatants of IFN-γ stimulated HBL-1 cells contained predominantly exPD-L1, production of which was blocked by GW4869, a neutral sphingomyelinase inhibitor that blocks exosome generation, or by siRNA-mediated knockdown of *HGS* (*Figure 3—figure*

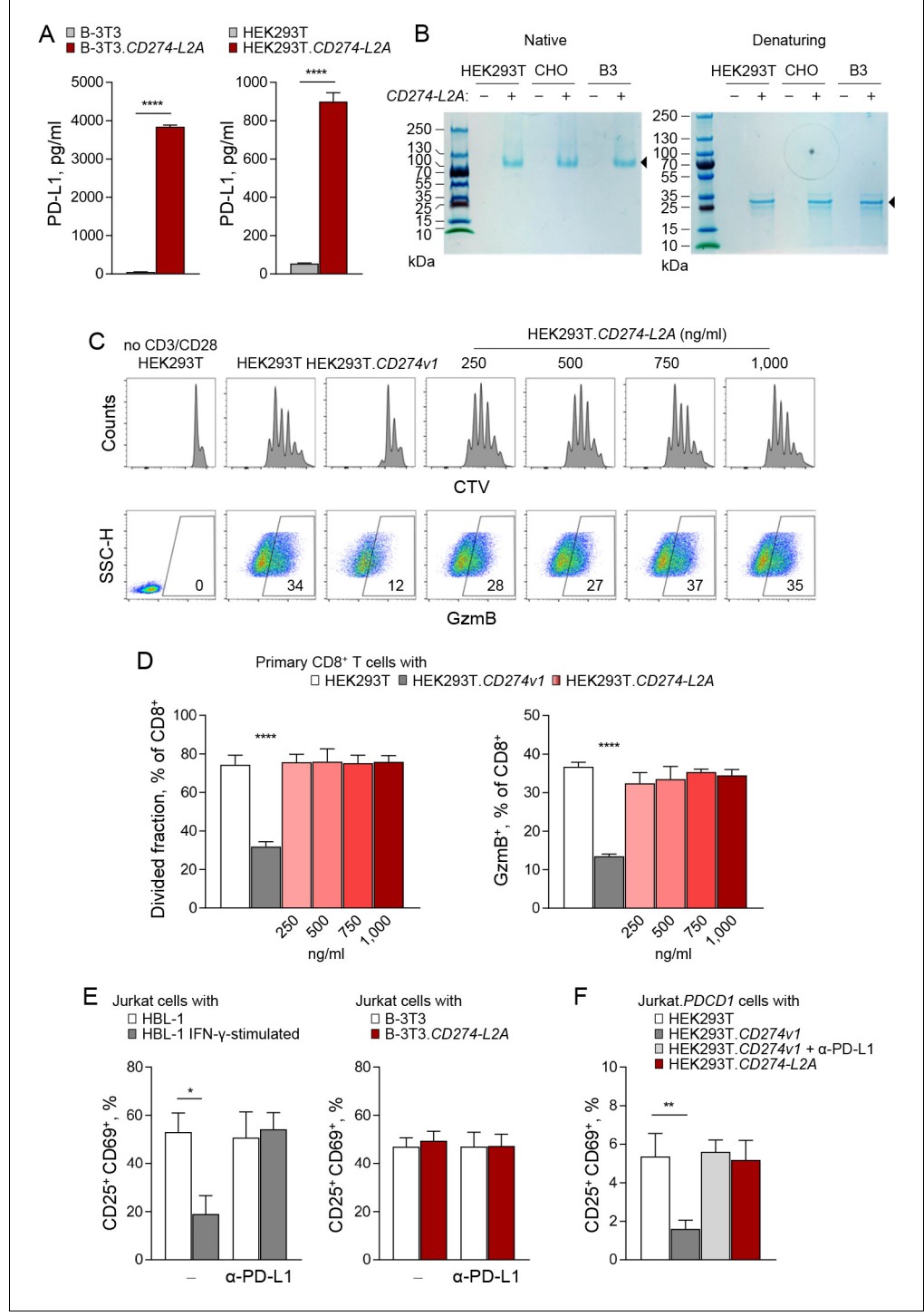

**Figure 4.** *CD274-L2A*-derived soluble PD-L1 is not immunosuppressive. (**A**) Quantification of soluble PD-L1 by ELISA in supernatants of B-3T3 and HEK293T cells retrovirally transduced with *CD274-L2A*. Mean (± SEM) concentration from three independent experiments are shown. (**B**) Coomassie Brilliant Blue stain, under native or reducing (β-ME/SDS) PAGE conditions, of serum-free supernatants from HEK293T, CHO and B3 cells transfected or not with *CD274-L2A*. (**C–D**) Primary CD8[+] T cells were labelled with CTV and stimulated with CD3- and CD28-coated beads for 72 hr, alone or co-cultured with HEK293T, HEK293T.*CD274v1* or HEK293T.*CD274-L2A* cells transfected with the indicated amount of plasmid DNA. T cells were stained for intracellular GzmB at the end of the culture period. Representative histograms and scatter plots are shown in C; quantification of CTV[lo] and GzmB[+] cells of three healthy donors according to the amount of transfected plasmid DNA is shown in D. (**E**) Percentage

*Figure 4 continued on next page*

*Figure 4 continued*

of activated (CD25$^+$CD69$^+$) Jurkat cells in the presence of conditioned media from parental HBL-1 cells, IFN-γ-stimulated HBL-1 cells, parental B-3T3 cells or B-3T3.*CD274-L2A* transduced cells. Cells were stimulated with CD3- and CD28-coated beads for 24 hr, with 10 µg/mL of a PD-L1-blocking antibody added where indicated. Mean (± SEM) proportion from three independent experiments are shown. (**F**) Percentage of activated (CD25$^+$CD69$^+$) Jurkat.*PDCD1* cells after co-cultured with parental HEK293T, HEK293T.*CD274v1*, or HEK293T.*CD274-L2A* cells. Cells were stimulated with CD3- and CD28-coated beads for 24 hr, with 10 µg/mL of a PD-L1-blocking antibody added where indicated. Mean (± SEM) proportion from three independent experiments are shown.
The online version of this article includes the following figure supplement(s) for figure 4:

**Figure supplement 1.** Generation of Jurkat.*PDCD1* cells.

*supplement 2*), which regulates endosomal transport and exosome formation (*Colombo et al., 2013*; *Sun et al., 2016*; *Tamai et al., 2010*). In contrast, sPD-L1 produced by HEK293T.*CD274-L2A* cells was not affected by these treatments (*Figure 3—figure supplement 2*). Moreover, sPD-L1 in the supernatant of *CD274-L2A*-transfected cells migrated close to its theoretical molecular weight of 28 kDa under reducing gel electrophoresis conditions, but appeared multimerised under native conditions, independently of the type of cell line, in which it was produced (*Figure 4B*). Multimerisation of sPD-L1 seen here is in agreement with recent findings (*Mahoney et al., 2019*), albeit the multimer in our study had a molecular weight consistent with a tetramer, rather than a dimer.

To test the potential immunosuppressive activity of *CD274-L2A*-derived sPD-L1, we activated primary CD8$^+$ T cells isolated from healthy donors with CD3- and CD28-coated beads in conditioned media from HEK293T.*CD274-L2A* or control HEK293T cells. Under these conditions, no effect of *CD274-L2A*-derived sPD-L1 could be measured on either proliferation or granzyme B (GzmB) production of stimulated CD8$^+$ T cells (*Figure 4C,D*). In contrast, incubation with HEK293T cells transduced with full-length PD-L1-encoding *CD274* variant 1 (HEK293T.*CD274v1*), expectedly and significantly suppressed CD8$^+$ T cell activation (*Figure 4C,D*).

To extend these observations, we used Jurkat T cells, which we activated with CD3- and CD28-coated beads. Again, *CD274-L2A*-derived sPD-L1 produced in B-3T3 cells had no measureable effect on Jurkat T cell expression of CD25 or CD69 (*Figure 4E*). In contrast, Jurkat T cell activation was considerably reduced by addition of conditioned media from HBL-1 cells in a PD-L1-dependent and exosome generation-dependent way (*Figure 4E*; *Figure 3—figure supplement 2*).

As Jurkat T cells express minimal amounts of PD-1 prior to activation, we increased the sensitivity of PD-L1-mediated suppression in this system by stably overexpressing PD-1 in these cells. Jurkat T cells retrovirally transduced with PD-1-encoding *PDCD1* (Jurkat.*PDCD1*) exhibited high levels of surface PD-1, assessed by flow cytometry, and correct membrane localisation, assessed by immunofluorescence (*Figure 4—figure supplement 1*). Co-culture with HEK293T.*CD274v1* displaying membrane-bound PD-L1 significantly reduced activation of Jurkat.*PDCD1* cells by CD3 and CD28 stimulation in a PD-L1-dependent manner (*Figure 4F*). Under the same conditions, incubation of Jurkat.*PDCD1* cells with *CD274-L2A*-derived sPD-L1 produced in HEK293T cells had no impact on their response to CD3 and CD28 stimulation (*Figure 4F*). Therefore, no suppressive activity could be demonstrated for the native *CD274-L2A* protein product under any of the conditions we have studied.

## *CD274-L2A* produces a natural PD-1 antagonist

Although the *CD274-L2A* protein product had no demonstrable suppressive activity in the assays employed here, it nevertheless contained intact PD-1-binding domains. We therefore considered the possibility that binding of *CD274-L2A*-derived sPD-L1 to PD-1 did not initiate suppressive signalling, but instead antagonised binding of the membrane-bound suppressive form of PD-L1.

To test this hypothesis directly, we activated healthy donor primary CD8$^+$ T cells with CD3 and CD28 stimulation and suppressed this activation with membrane-bound PD-L1 provided by HEK293T.*CD274v1* cells (*Figure 5A*). As expected, this PD-L1-mediated suppression was fully restored upon addition of anti-PD-L1 antibodies (*Figure 5A*). Surprisingly, addition of conditioned media from HEK293T.*CD274-L2A* was also able to fully restore responsiveness of primary CD8$^+$ T cells to CD3 and CD28 stimulation (*Figure 5A*). These data suggested that *CD274-L2A*-derived sPD-

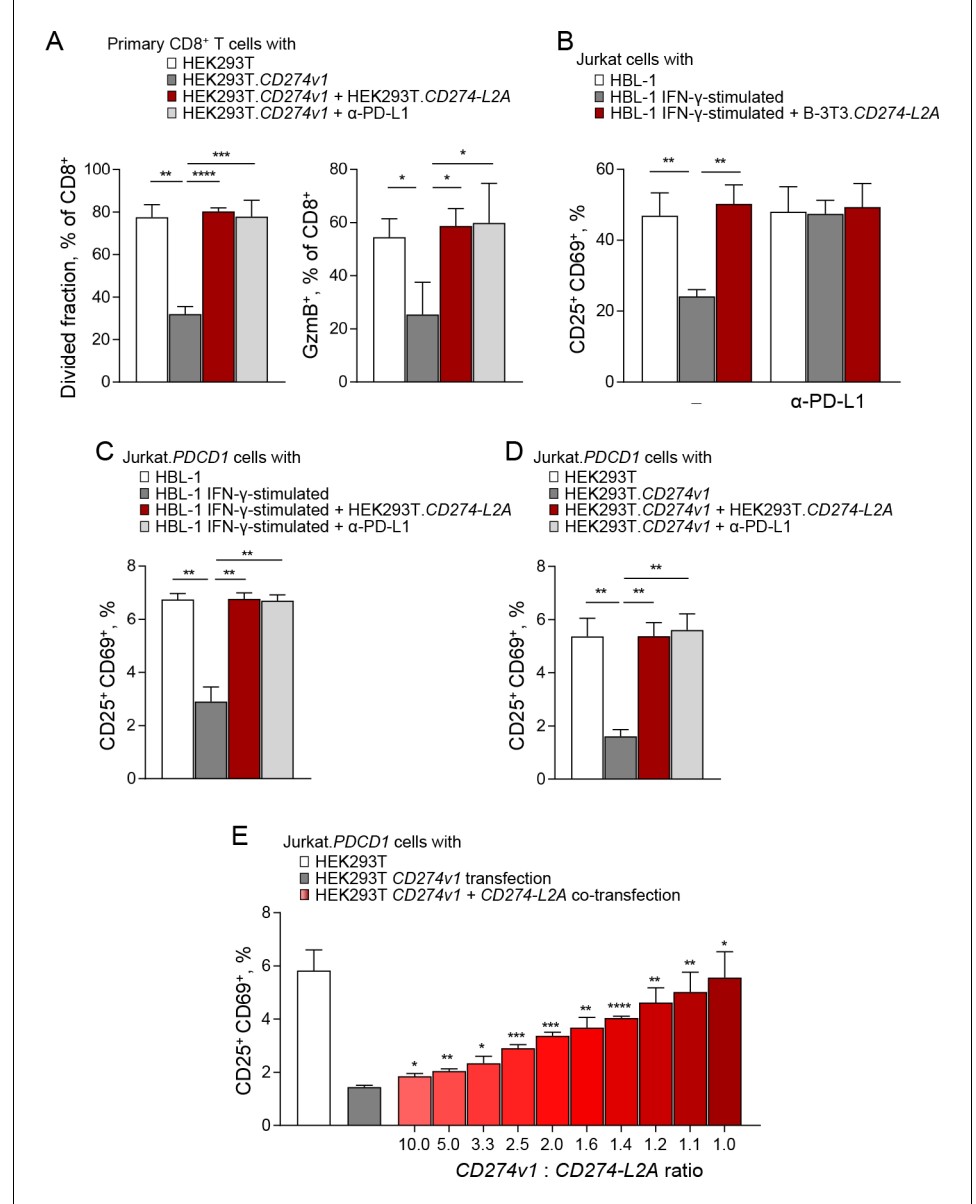

**Figure 5.** *CD274-L2A*-derived soluble PD-L1 acts as a receptor antagonist in the presence of transmembrane PD-L1. (**A**) Primary CD8[+] T cells were labelled with Cell Trace Violet (CTV) and stimulated with CD3- and CD28-coated beads for 72 hr. T cells were co-cultured with parental HEK293T or HEK293T.*CD274v1* transfected cells in the presence of conditioned media from HEK293T.*CD274-L2A* transduced cells or a PD-L1-blocking antibody. Mean (± SEM) proportion of CTV[lo] and GzmB[+] cells of three healthy donors is shown. (**B**) Percentage of activated (CD25[+]CD69[+]) Jurkat cells in the presence of conditioned media from parental HBL-1 cells, IFN-γ-stimulated HBL-1 cells, and transduced B-3T3.*CD274-L2A* cells. Cells were stimulated with CD3- and CD28-coated beads for 24 hr, with 10 μg/mL of a PD-L1-blocking antibody added where indicated. Mean (± SEM) proportion from three independent experiments are shown. (**C**) Percentage of activated (CD25[+]CD69[+]) Jurkat.*PDCD1* cells in the presence of conditioned media from parental HBL-1 cells, IFN-γ-stimulated HBL-1 cells, and transduced HEK293T.*CD274-L2A* cells. Cells were stimulated with CD3- and CD28-coated beads for 24 hr, with 10 μg/mL of a PD-L1-blocking antibody added where indicated. Mean (± SEM) proportion from three independent experiments are shown. (**D**) Percentage of activated (CD25[+]CD69[+]) Jurkat.*PDCD1* cells following co-culture with parental HEK293T or HEK293T.*CD274v1* transfected cells in the presence of conditioned media from HEK293T.*CD274-L2A* or a PD-L1-blocking antibody. Cells were stimulated with CD3- and CD28-coated beads for 24 hr. Mean (± SEM) proportion from three independent experiments are shown. (**E**) Percentage of activated (CD25[+]CD69[+]) Jurkat.*PDCD1* cells following co-culture with HEK293T cells transfected with varying ratios of *CD274v1* and *CD274-L2A* as shown. The concentration of the *CD274v1* plasmid is kept constant across all conditions at 1000 ng. Cells were

*Figure 5 continued on next page*

*Figure 5 continued*
stimulated with CD3- and CD28-coated beads for 24 hr. Mean (± SEM) proportion from three independent experiments are shown.
The online version of this article includes the following figure supplement(s) for figure 5:
**Figure supplement 1.** *CD274v1* and *CD274-L2A* predominantly produce transmembrane and soluble PD-L1 respectively.
**Figure supplement 2.** Surface PD-L1 is not lost upon co-transfection with *CD274-L2A*.

L1 acted as a PD-1 antagonist, blocking the suppressive action of membrane-bound PD-L1 as efficiently as an anti-PD-L1 antibody.

These observations were repeated when the response of Jurkat T cells to CD3 and CD28 stimulation was suppressed by exPD-L1 produced by IFN-γ stimulated HBL-1 cells (*Figure 5B*). This activity in the supernatant of HBL-1 cells depended on exosome generation (*Figure 3—figure supplement 2*) and was blocked by anti-PD-L1 antibodies (*Figure 5B*). Again, the addition of *CD274-L2A*-derived sPD-L1, produced in B-3T3 cells fully restored the response of Jurkat T cells to CD3 and CD28 stimulation in the presence of HBL-1-produced exPD-L1 (*Figure 5B*). Comparable results were also obtained when HBL-1-produced exPD-L1 was used to suppress the response of Jurkat.*PDCD1* cells to CD3 and CD28 stimulation, which again was restored by *CD274-L2A*-derived sPD-L1, produced in HEK293T cells, as efficiently as by an anti-PD-L1 antibody (*Figure 5C*). Moreover, *CD274-L2A*-derived sPD-L1, produced in HEK293T cells, was also able to block the suppression of CD3 and CD28 stimulated Jurkat.*PDCD1* cells by membrane-bound PD-L1 provided by HEK293T.*CD274v1* cells (*Figure 5D*).

These experiments demonstrated that the *CD274-L2A* protein product displayed PD-1 antagonism, blocking the suppressive activity of full-length PD-L1 bound on the membranes of intact cells or exosomes. However, the relative ratios of membrane-bound PD-L1 and *CD274-L2A*-derived sPD-L1 in these assays are likely to differ from those suggested by the RNA-seq data analysis. To examine whether *CD274-L2A*-derived sPD-L1 can block the suppressive activity of membrane-bound PD-L1 at defined physiological ratios of the two forms, we used a co-expression approach. We constructed expression plasmids expressing either the *CD274* variant 1 (*CD274v1*) or *CD274-L2A*, which we transfected into HEK293T cells. Transfection with *CD274v1* alone increased the surface levels of PD-L1 in a dose-dependent manner, without generating detectable sPD-L1 (*Figure 5—figure supplement 1*), arguing against proteolytic processing of the full-length PD-L1 into smaller, soluble products. In contrast, transfection with *CD274-L2A* alone increased, in a dose-dependent manner, the levels of sPD-L1 detected by ELISA in the supernatant of HEK293T cells, without any increase in surface PD-L1 (*Figure 5—figure supplement 1*). We then co-transfected the two plasmids at different ratios. Of note, co-transfection with *CD274v1* did not affect the surface expression of PD-L1 derived from *CD274v1* transfection (*Figure 5—figure supplement 2*). Importantly, co-transfection with *CD274-L2A* reversed the suppressive effect of *CD274v1*-derived membrane-bound PD-L1 when HEK293T cells were co-cultured with CD3 and CD28 stimulated Jurkat.*PDCD1* cells (*Figure 5E*). Collectively, these demonstrate that the *CD274-L2A* protein product is a natural PD-1 antagonist, which allows T cells to overcome PD-L1-mediated suppression and achieve maximal activation.

In order to examine the biological activity for sPD-L1 in vivo, we used a murine tumour model, based on transplantation of MCA-38 colon adenocarcinoma cells, which is responsive to PD-1 or PD-L1 blockade (*Gong et al., 2019*). As murine MCA-38 cells do not produce sPD-L1 (*Figure 2—figure supplement 1B*), we transduced them to produce either human sPD-L1, encoded by human *CD274-L2A*, or a chimeric sPD-L1, consisting of the murine sequence encoded by the murine exons 1–4, followed by the human 18 C-terminal amino acids, encoded by the retained part of human intron 4 and *L2A* element. The human sPD-L1 was chosen on the basis of its reported ability to bind murine PD-1, also in the context of the MCA-38 tumour model (*Fenwick et al., 2019*; *Huang et al., 2017*). Transduced MCA-38 cell lines produced murine or human sPD-L1, detectable in cell supernatant by species-specific ELISAs, respectively, and exhibited in vitro growth kinetics comparable to MCA-38 cells transduced with a GFP-encoding vector (*Figure 6—figure supplement 1A,B*). When subcutaneously transplanted into immunocompetent mice, expression of either murine or human sPD-L1 significantly delayed the growth kinetics of MCA-38 tumours, which was reflected in the tumour masses at the

end of the observation period (*Figure 6A,B*). This effect on tumour growth was comparable to the effect of anti-PD-L1 blockade (*Figure 6A,B*), indicating in vivo activity of sPD-L1 as an antagonist of the PD-1 – PD-L1 axis.

## Discussion

EREs represent a dynamic source of genetic diversity, providing substrates for the evolution of novel host functions, including those of immune genes (*Burns and Boeke, 2012*; *Feschotte and Gilbert, 2012*; *Kassiotis and Stoye, 2016*). For example, a HERV integration upstream of the *CD5* gene initiates an alternative transcript in healthy B cells that omits the signal peptide-encoding first exon of the canonical transcript, resulting in an intracellularly retained form of the otherwise transmembrane CD5 protein (*Renaudineau et al., 2005*). Differential initiation by the HERV integration or the canonical promoter therefore regulates transmembrane CD5 expression (*Renaudineau et al., 2005*).

Here, we provided another example where ERE exonisation truncates a transmembrane immune protein, thereby regulating its function. Exonisation of an intronic *L2A* element generates a 3' truncated *CD274* transcript, omitting the transmembrane domain-encoding sequence and producing secreted sPD-L1. Secreted forms of sPD-L1 with both Ig-like domains or only the Ig-like V-type domain intact can also be produced by *CD274* splice variant 9 and variants 3/12, respectively. However, these variants were infrequently expressed in our sample collection and were also comparably infrequently expressed in independently analysed cohorts (*Gong et al., 2019*). Moreover, cell-free forms of PD-L1 were not detected in cells overexpressing exclusively the full-length *CD274* transcript, arguing against proteolytic cleavage as a major source of sPD-L1. Together, these observations indicate *CD274-L2A* as the predominant source of sPD-L1, at least in the majority of tumour and healthy samples.

Supporting *CD274-L2A* as the main source of sPD-L1 in humans is the observation that laboratory mice, in which sPD-L1 has not been described, have lost the ability to generate the equivalent *Cd274-L2A* transcript. The features necessary to generate the *CD274-L2A* variant are remarkably evolutionarily conserved in humans and all other hominids. In stark contrast, these features seem to have taken a different evolutionary trajectory in the common ancestor of mice, rats, and hamsters, where there has been apparent selection to prevent exonisation of the intronic *L2A* element by strengthening the canonical splice site and removing the polyadenylation signal in this element.

The exaptation of *L2A* element in the generation of sPD-L1 in humans likely extends beyond the provision of an alternative terminal exon, omitting the transmembrane domain. In contrast to all

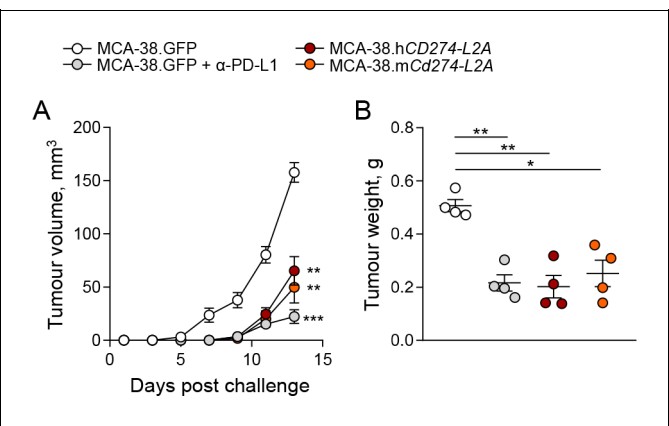

**Figure 6.** *CD274-L2a*-derived soluble PD-L1 delays in vivo tumour growth. (**A–B**) Tumour growth following subcutaneous inoculation of $1 \times 10^6$ MCA-38 cells expressing human sPD-L1 (MCA-38.h*CD274-L2A*), a constructed murine sPD-L1 variant (MCA-38.m*Cd274-L2A*) or GFP. One group of recipient mice was treated with anti-PD-L1 antibodies. Mean tumour volumes (± SEM) throughout the experiment (**A**) and tumour weights at endpoint (**B**) of 4 mice per group are plotted from one representative of two experiments.

The online version of this article includes the following figure supplement(s) for figure 6:

**Figure supplement 1.** Characterisation of sPD-L1-producing MCA-38 cells.

other splice variants that could also produce sPD-L1, *CD274-L2A* also omits the canonical 3' UTR, with important consequences for its regulation. PD-L1 is constitutively expressed in a wide variety of hematopoietic and non-hematopoietic cells and can be strongly induced by IFNs and other inflammatory cytokines in autoimmunity, infection, or cancer (*Sharpe and Pauken, 2018*). Balancing positive regulation by inflammatory stimuli, PD-L1 production is controlled post-transcriptionally by regulatory elements the full-length *CD274* mRNA (*Coelho et al., 2017*). Indeed, binding of tristetraprolin (TTP) to AU-rich elements (ARE) in 3' UTR of the *CD274* mRNA decreases its stability, thereby reducing surface PD-L1 expression, a mechanism that is counteracted by oncogenic RAS mutations (*Coelho et al., 2017*). The importance of this regulatory region is also illustrated in frequent *CD274* 3' UTR structural variants found in many cancers, causing elevated surface PD-L1 expression in cancer cells (*Kataoka et al., 2016*). Lack of an ARE in *CD274-L2A*, as well as in the other two *CD274* variants described here, which use a terminal ERE instead on the canonical 3'UTR, would render them resistant to the destabilising effect of TTP. Differential post-transcriptional regulation according to the presence of the ARE could, in principle, explain the often disparate expression of full-length *CD274* and *CD274-L2A* in different types of healthy and transformed cells. Alternatively, differential full-length *CD274* and *CD274-L2A* expression might result from the activity of a specific splicing factor, identification of which will require further investigation.

The critical immune suppressive role for full-length PD-L1, bound on the membrane of cells or exosomes, is amply demonstrated both in animal models, where genetic deficiency or antibody blockade results in severe phenotypes, and in the success of cancer immunotherapy based on blockade of the PD-L1 – PD-1 interaction (*Chamoto et al., 2017*; *Chen et al., 2018*; *Francisco et al., 2010*; *Sharpe and Pauken, 2018*; *Sun et al., 2018*). However, the biological activity of membrane-free sPD-L1 had remained puzzling.

Early studies using recombinant sPD-L1 suggested an immune suppressive role either in inhibiting T cell activation or in promoting T cell apoptosis (*Chinnadurai et al., 2014*; *Frigola et al., 2011*). More recent studies using recombinant proteins encoded by *CD274-L2A* (*Hassounah et al., 2019*; *Mahoney et al., 2019*) or variants 9 and 3/12 (*Zhou et al., 2017*), also suggested a T cell inhibitory activity for sPD-L1, comparable to or higher than that of the full-length protein. However, the reported activity might not reflect that of the natural sPD-L1 protein, but rather of the artificial Fc fusion that was used in the majority of these studies (*Chinnadurai et al., 2014*; *Frigola et al., 2011*; *Hassounah et al., 2019*; *Zhou et al., 2017*). Indeed, fusion to Fc is likely to affect the valency of sPD-L1 and may even be presented in a membrane-bound from through binding to Fc receptors. In some of these studies, the sPD-L1-Fc fusion was presented in a plate-bound form (*Chinnadurai et al., 2014*), which again would not reflect the soluble nature of the natural molecule. The commercially available recombinant sPD-L1-Fc fusions used in some of these studies are marketed as suppressive molecules. Nevertheless, it is interesting to note that at least two reports using the same sPD-L1-Fc fusions found that they can promote, rather than suppress, T cell activation in the presence of APCs (*Steidl et al., 2011*; *Wan et al., 2006*) – an effect that is likely explained by receptor antagonism.

Mahoney et al., recently demonstrated that human sPD-L1, encoded by *CD274-L2A* and produced by transfection of murine cells, naturally forms dimers and higher-order multimers and that a His-tagged recombinant version, when used at 10–20 μg/ml, can suppress T cell activation in response to CD3 stimulation (*Mahoney et al., 2019*). We also observed multimeric forms of sPD-L1 produced in human cells, although these were consistent with tetramers, rather than dimers. Intrinsic disparities in producing cell lines (e.g. glycosylation patterns), might account for the observed differences in molecular size of any multimer or, indeed, the monomer, which appeared smaller and closer to the theoretical size of 28 kDa in our study. Nevertheless, in none of the multiple conditions, we tested did we observe suppression of T cell activation, using more physiological concentrations (1–2 ng/ml) of sPD-L1 produced in murine or human cells. Similarly, no PD-L1-mediated suppressive activity could be demonstrated by independent studies of cell-free, non-exosomal fractions (*Chen et al., 2018*), or of the natural product of *CD274* variant 9, even at 2 μg/ml (*Gong et al., 2019*).

The consensus from these studies is that *CD274-L2A*-encoded sPD-L1, whilst retaining PD-1 binding activity, lacks suppressive activity. Such properties are consistent with a receptor antagonist role and, indeed, our data clearly demonstrate the ability of *CD274-L2A*-encoded sPD-L1 to reverse T cell suppression mediated by membrane-bound PD-L1, both cellular and exosomal. The generation of receptor antagonist sPD-L1 by alternative splicing is not only at the expense of the canonical

transcript encoding the agonist; it additionally offers a means of producing the antagonist without first producing the agonist, as would be the case with proteolytic cleavage.

Antagonistic activity by sPD-L1 is reminiscent of a growing number of membrane-bound cytokines as well as co-stimulatory or co-inhibitory molecules and their receptors, whose soluble versions assume a blocking or antagonist function (*Bemelmans et al., 2017*; *Dinarello, 1998*; *Gu et al., 2018*; *Guégan and Legembre, 2018*; *Zhu and Lang, 2017*). Although soluble forms of many of these molecules are thought to be produced by proteolytic cleavage of the membrane-bound forms, many others are produced by alternatively spliced mRNAs (*Gu et al., 2018*). These include soluble PD-1, soluble CTLA-4, and soluble CD80, all of which are produced by alternative transcript variants that omit their respective transmembrane domain-encoding exons (*Kakoulidou et al., 2007*; *Magistrelli et al., 1999*; *Nielsen et al., 2005*). Generation of soluble forms of several co-inhibitory molecules underscores their broader involvement in immune regulation, but also the complexity of their interconnected pathways. Whilst certainly not acting alone, the evolutionary conservation of sPD-L1 production in humans suggests it is an important contributor to the regulation of PD-L1 activity.

Membrane-bound PD-L1 can be antagonised also by in cis binding to CD80, both in human and murine dendritic cells (*Sugiura et al., 2019*), and sPD-1 has long been hypothesised to function as a decoy receptor, reducing availability of membrane-bound PD-L1 (*Zhu and Lang, 2017*). Moreover, sPD-L1 can interfere with the effect of anti-PD-L1 immunotherapy. Indeed, tumours harbouring mutations in the TDP-43 splicing factor from two non-small cell lung cancer patients have recently been shown to overproduce sPD-L1 encoded by *CD274* variant 9 or an alternative splice variant omitting exons 5 and 6, which can act as a decoy for PD-L1-targeting antibodies, thereby reducing the effect of immunotherapy (*Gong et al., 2019*).

Complex and context-dependent regulation would be expected for a regulatory pathway as important as the PD-L1 – PD-1 axis, the individual components of which are still emerging. Their careful dissection in new animal models will illuminate both their relative contribution to the regulation of the pathway, as well as the distinct evolutionary trajectories that can achieve the same outcome.

## Materials and methods

### Key resources table

| Reagent type (species) or resource | Designation | Source or reference | Identifiers | Additional information |
|---|---|---|---|---|
| Genetic reagent (*Mus musculus*) | C57BL/6J | The Jackson Laboratory | RRID: IMSR_JAX:000664 | |
| Cell line (*Homo sapiens*) | HEK293T | Cell Services Facility, Francis Crick Institute | RRID: CVCL_0063 | |
| Cell line (*Homo sapiens*) | Jurkat | Cell Services Facility, Francis Crick Institute | RRID: CVCL_0065 | |
| Cell line (*Chlorocrbus sabaeus*) | Vero | Cell Services Facility, Francis Crick Institute | RRID: CVCL_0059 | |
| Cell line (*Chlorocebus aethiops*) | CV-1 | Cell Services Facility, Francis Crick Institute | RRID: CVCL_0229 | |
| Cell line (*Oryctolagus cuniculus*) | R9ab | Cell Services Facility, Francis Crick Institute | RRID: CVCL_3782 | |
| Cell line (*Cricetulus griseus*) | CHO | Cell Services Facility, Francis Crick Institute | RRID: CVCL_0213 | |
| Cell line (*Mus musculus*) | B-3T3 | Cell Services Facility, Francis Crick Institute | RRID: CCL-163 | |
| Cell line (*Mus musculus*) | EL4 | Cell Services Facility, Francis Crick Institute | RRID: CVCL_0255 | |
| Cell line (*Mus musculus*) | MCA-38 | Cell Services Facility, Francis Crick Institute | RRID: CVCL_B288 | |

*Continued on next page*

*Continued*

| Reagent type (species) or resource | Designation | Source or reference | Identifiers | Additional information |
|---|---|---|---|---|
| Cell line (*Mus musculus*) | B3 | Cell Services Facility, Francis Crick Institute | RRID: CVCL_RP56 | |
| Antibody | Rat monoclonal anti-mouse PD-L1 (clone 10F.9G2) | Biolegend (FACS), BioXCell (in vivo) | Cat: 124315 (Biolegend) Cat: BE0101 (BioXCell) | FACS (1:200) In vivo injection (200 ug i.p.) |
| Antibody | Mouse monoclonal anti-human PD-L1 (clone 29E.2A3) | Biolegend (FACS) | Cat: 329706 | FACS (1:200) |
| Antibody | Mouse monoclonal anti-human PD-1 (clone EH12.2H7) | Biolegend (FACS) | Cat: 329908 | FACS (1:200) |
| Antibody | Mouse monoclonal anti-human CD25 (clone BC96) | Biolegend (FACS) | Cat: 302642 | FACS (1:200) |
| Antibody | Mouse monoclonal anti-human CD69 (clone FN50) | Biolegend (FACS) | Cat: 310906 | FACS (1:200) |
| Antibody | Mouse monoclonal anti-human CD8 (clone SK1) | Biolegend (FACS) | Cat: 344710 | FACS (1:200) |
| Antibody | Mouse monoclonal anti-human Granzyme B (clone QA16A02) | Biolegend (FACS) | Cat: 372204 | FACS (1:200) |
| Antibody | Mouse monoclonal anti-human HGS (clone C-7) | Santa Cruz (WB) | Cat: sc-271455 | WB (1:1000) |
| Antibody | HRP-conjugated mouse monoclonal anti-human Actin (clone AC-15) | Abcam (WB) | Cat: ab49900 | WB (1:25000) |
| Chemical compound, drug | GW4869 | Sigma Aldrich | Cat: D1692 | |
| Transfected construct (*Homo sapiens*, *Mus musculus*) | pRV-IRES-GFP (lentiviral vector) | This paper | | Lentiviral construct expressing GFP; used as empty vector control |
| Transfected construct (*Homo sapiens*, *Mus musculus*) | pRV-CD274-L2A-IRES-GFP (lentiviral vector) | This paper | | Lentiviral construct expressing human *CD274-L2A* |
| Transfected construct (*Mus musculus*) | pRV-mCd274-L2A-IRES-GFP (lentiviral vector) | This paper | | Lentiviral construct expressing murine-human chimeric *Cd274-L2A* |
| Transfected construct (*Homo sapiens*) | pRV-PDCD1-IRES-GFP (lentiviral vector) | This paper | | Lentiviral construct expressing human *PDCD1* (PD-1) |
| Recombinant DNA reagent | pcDNA3.1-CD274 (plasmid) | This paper | | Mammalian expression plasmid encoding human *CD274* |
| Recombinant DNA reagent | pcDNA3.1-CD274-L2A (plasmid) | This paper | | Mammalian expression plasmid encoding human *CD274-L2A* |
| Recombinant DNA reagent | pcDNA3.1-CD274 IS4 (plasmid) | This paper | | Mammalian expression plasmid encoding human *CD274* with intron 4 |
| Recombinant DNA reagent | pcDNA3.1-CD274 IS4ΔL2A (plasmid) | This paper | | Mammalian expression plasmid encoding human *CD274* with intron four with *L2A* sequence deleted |

*Continued on next page*

*Continued*

| Reagent type (species) or resource | Designation | Source or reference | Identifiers | Additional information |
|---|---|---|---|---|
| Commercial assay or kit | Human PD-L1 ELISA kit | Abcam | Cat: ab214565 | |
| Commercial assay or kit | Mouse PD-L1 ELISA kit | Biomatik | Cat: EKU06803 | |
| Commercial assay or kit | RNeasy Mini RNA extraction kit | Qiagen | Cat: 74104 | |
| Commercial assay or kit | High Capacity cDNA Reverse Transcription kit | Applied Biosystems | Cat: 4368814 | |

## Mice and tumour challenge

Inbred C57BL/6J (B6) mice were originally obtained from The Jackson Laboratory and subsequently maintained at the Francis Crick Institute's animal facilities. Eight- to 12-week-old male mice were used for all experiments, randomly assigned to the different treatment groups. All animal experiments were approved by the ethical committee of the Francis Crick Institute and conducted according to local guidelines and UK Home Office regulations under the Animals Scientific Procedures Act 1986 (ASPA) (licence number: PCD77C6D0). For tumour studies, $1 \times 10^6$ MCA-38 derivative cell lines were subcutaneously inoculated into the right flank of recipient mice. Where indicated, mice received intraperitoneal (i.p.) injections of 200 µg anti-PD-L1 (clone 10F.9G2; BioXCell) on days 1, 3, 5, and 7 after tumour inoculation.

## Transcript assembly and repeat region annotation

Transcripts were assembled using RNA-seq reads as previously described (*Attig et al., 2019*). Briefly, RNA-seq reads from 768 cancer patient samples obtained from The Cancer Genome Atlas (TCGA) program were used to generate a pan-cancer transcriptome by Trinity (*Grabherr et al., 2011*) (v2.2.0). The transcript assembly was then annotated against GENCODE (basic, version 24) (*Frankish et al., 2019*). Hidden Markov models (HMMs), representing known Human repeat families (Dfam 2.0 library v150923) were used to annotate GRCh38 using RepeatMasker (www.repeatmasker.org), configured with nhmmer (*Wheeler and Eddy, 2013*), as previously described (*Attig et al., 2017*). RepeatMasker annotates LTR and internal regions separately, thus tabular outputs were parsed to merge adjacent annotations for the same element.

## Read mapping and counting

RNA-seq reads were obtained from TCGA, the Genotype-Tissue Expression (GTEx) program, the Cancer Cell Line Encyclopedia (CCLE), and the indicated individual studies and aligned to our custom transcript assembly, as previously described (*Attig et al., 2019*). Alternatively, reads were aligned to the GENCODE (basic, version 30), with the *CD274-L2A* transcript replacing the partially overlapping ENST00000474218 transcript. TPM calculations were carried out for all transcripts with a custom Bash pipeline using GNU parallel (*Tange, 2011*) and Salmon (v0.8.2 or v0.9.2) (*Patro et al., 2017*). It should be noted that TPM values calculated here using our custom transcript assembly are on average four times lower than those calculated using the previously annotated transcriptome, since an integral part of TPM calculation is division by the total transcript count, which is substantially higher in the former, than in the latter. Splice junction analysis was carried out using the integrated function of the Integrative Genome Viewer (IGV v2.4.19, the Broad Institute). Downstream analysis and visualization was conducted with Qlucore Omics Explorer (Qlucore, Lund, Sweden).

## Sequence alignments

Multiple genomic sequence alignments were carried out using the comparative genomic tool from Ensembl (https://www.ensembl.org/info/genome/compara/multiple_genome_alignments.html) and with Vector NTI (11.5.0). Sequences were downloaded and plotted with Vector NTI or with AliView

(1.21) (*Larsson, 2014*). For sequence divergence of *CD274* introns and exons the genomic sequences from the following 10 primate species were compared: *Chlorocebus sabaeus*, *Gorilla gorilla*, *Homo sapiens*, *Macaca fascicularis*, *Macaca mulatta*, *Pan paniscus*, *Pan troglodytes*, *Papio Anubis*, *Pongo abelii* and *Theropithecus gelada*.

## Primary cells

Peripheral blood was collected from healthy adult volunteers according to protocols approved by the ethics board of The Francis Crick Institute. Peripheral blood mononuclear cells (PBMCs) were freshly isolated by density gradient centrifugation in Ficoll-Paque (VWR) and CD8$^+$ T cells isolated by negative selection using a CD8$^+$ T Cell Isolation Kit (Miltenyi Biotec).

## Cell lines

HEK293T, Jurkat, B-3T3, EL4, MCA-38, CHO, B3, Vero, CV-1 and R9ab cells were obtained from and verified as mycoplasma free by the Cell Services facility at The Francis Crick Institute. Human cell lines were also validated by DNA fingerprinting. All cells were grown in Iscove's Modified Dulbecco's Medium (Sigma-Aldrich) supplemented with 5% fetal bovine serum (Thermo Fisher Scientific), L-glutamine (2 mmol/L, Thermo Fisher Scientific), penicillin (100 U/mL, Thermo Fisher Scientific), and streptomycin (0.1 mg/mL, Thermo Fisher Scientific). HEK293T, B-3T3, MCA-38, B3 and Jurkat sublines transduced with *CD274v1*, *CD274-L2A*, or *PDCD1* were generated by adding viral stocks (as described above) to target cells in the presence of polybrene (4 µg/mL) and sorted based on GFP expression to >98% purity on a S3e Cell Sorter (Propel Labs Inc), 72 hr after transduction. Supernatant PD-L1 levels were measured 48 hr after seeding $5 \times 10^5$ sorted cells/ml, using the human PD-L1 ELISA Kit (clone 28–8, Abcam) or the murine PD-L1 ELISA Kit (EKU06803, Biomatik), according to manufacturers' instructions.

## Retroviral and expression vectors

Open-reading frames encoding truncated human or murine-human chimeric *CD274-L2A*, and human *PDCD1* were synthesized and cloned into the pRV-GFP vector, constructed and kindly provided by Dr. Gitta Stockinger (The Francis Crick Institute, London, UK). Gene synthesis, cloning and mutagenesis were performed by Genewiz LLC and verified by sequencing. Vesicular stomatitis virus glycoprotein (VSVg)-pseudotyped retroviral particles were produced by transfection of HEK293T cells using GeneJuice (EMD Millipore) of vector plasmids with packaging (pHIT60) and VSVg (pcVG-wt) plasmids, kindly provided by Dr. Jonathan Stoye (The Francis Crick Institute, London, UK). Virus-containing supernatants were collected 48 hr post-transfection, passed through a 0.45 µm filter and stored at −80°C until further use. Open-reading frames encoding human full-length *CD274* or truncated *CD274-L2A*, and minigenes comprising the full-length *CD274* cDNA with an intact intron 4 (*CD274* IS4) or an intron 4 lacking the *L2A* element (*CD274* IS4ΔL2A), were additionally synthesized and cloned into the pcDNA3.1 mammalian expression vector for transient transfection experiments.

## Flow cytometry

Single-cell suspensions were stained for 30 min at room temperature with directly-conjugated antibodies to surface markers. For detection of intracellular antigens subsequent to surface staining, cells were fixed and permeabilised using the Foxp3/Transcription Factor Staining Buffer Set (Thermo Fisher Scientific) according to the manufacturer's instructions. The following antibodies were used: BV421-labelled mouse PD-L1 (clone 10F.9G2; Biolegend), PE-labelled human PD-L1 (clone 29E.2A3; Biolegend), APC-labelled human PD-1 (EH12.2H7; Biolegend), PE-Cy5-labelled or APC Fire 750-labelled human CD25 (BC96; Biolegend), PE-labelled human CD69 (FN50; Biolegend), APC Annexin V (Biolegend), PerCP-Cy5.5-labelled human CD8 (SK1; Biolegend), and APC-labelled human Granzyme B (QA16A02; Biolegend). Multicolour cytometry data were acquired on a LSR Fortessa (BD Biosciences) running BD FACSDiva v8.0 and analysed with FlowJo v10 (Tree Star Inc) analysis software.

## T cell suppression assays

CD8$^+$ T cells were labelled with Cell Trace Violet (CTV) (Thermo Fisher Scientific) for 20 min and rested overnight before being stimulated for 72 hr with CD3/CD28 Dynabeads (Thermo Fisher Scientific). Jurkat.*PDCD1* cells were stimulated for 24 hr with CD3/CD28 Dynabeads (Thermo Fisher

Scientific). Primary or cell line T cells were co-cultured with HEK293T cells transfected with pcDNA3.1-*CD274v1* or pcDNA3.1-*CD274-L2A* as indicated. Conditioned media from HEK293T. *CD274-L2A* cells, parental HEK293T cells, B-3T3.*CD274-L2A* cells, parental B-3T3 cells, IFN-γ-stimulated HBL-1 cells, parental HBL-1 cells, or 10 µg/mL ultra-LEAF anti-human PD-L1 antibody (29E.2A3, Biolegend) were added as indicated.

## Exosome inhibition

$5 \times 10^5$ HEK293T.*CD274-L2A* cells were grown in the presence of 10 µM GW4869 (Sigma-Aldrich) or DMSO control, or were transfected with varying concentrations of *HGS* siRNA (Insight Biotechnology). Supernatant was collected after 48 hr and PD-L1 quantified by ELISA according to the manufacturer's instructions (Abcam). *HGS* knockdown was assessed by western blot using anti-human HGS monoclonal antibody (Insight Biotechnology) and HRP-conjugated anti-beta actin (Abcam).

## Fluorescence microscopy

Cell lines were fixed in 4% paraformaldehyde and stained for 30 min at room temperature with PD-1 primary antibody (EH12.2H7, Biolegend) followed by 1 hr at room temperature with goat anti-mouse Alexa Fluor 647 secondary antibody (Thermo Fisher Scientific) and DAPI (Thermo Fisher Scientific). Cells were mounted in VectaShield mounting medium (Agilent) and imaged on an inverted LSM880 confocal microscope (Carl Zeiss AG).

## In vitro proliferation

$6 \times 10^3$ cells were seeded in triplicate in a flat-bottom 96-well plate and phase contrast microscopy was used to monitor cell growth on the IncuCyte S3 imaging system (EssenBioScience). Images were collected every 3 hr for 72 hr, and cell proliferation measured using the confluence image mask.

## Quantitative reverse transcriptase-based PCR (qRT-PCR)

$2 \times 10^6$ cells were stimulated with 100 ng/mL IFN-α or IFN-γ (Abcam) for 48 hr or used untreated. Total RNA from cell lines was isolated using the QIAcube (Qiagen), and cDNA synthesis was carried out with the High Capacity Reverse Transcription Kit (Applied Biosystems) with an added RNase inhibitor (Promega). Purified cDNA was used to quantify *CD274v1* and *CD274-L2A* using variant-specific primers.

For both variants in all species, the following common forward primer located in a conserved region of exon four was used:

Common primer | F: TACAGCTGAATTGGTCATCCCA.

The variant-specific reverse primers for human variants were:

*CD274v1* | R: TCAGTGCTACACCAAGGCAT

*CD274-L2A* | R: AGGCAGACATCATGCTAGGTG

The variant-specific reverse primers for African green monkey variants were:

*CD274v1* | R: TCAGTGCTACACCAAGGAGT

*CD274-L2A* | R: AGGCAGACATCATGCTAGGTG

The variant-specific reverse primers for leporine variants were:

*CD274v1* | R: TGAATGCTACACCAAGGAAC

*CD274-L2A* | R: ATGATGAATATTTACTAGGTG

The variant-specific reverse primers for murine variants were:

*CD274v1* | R: TGGACACTACAATGAGGAAC

*CD274-L2A* | R: GGATGGCCATCGTGCTAGGAA

For amplification of a conserved house-keeping gene, the following *HPRT*-specific primers were used in all species:

*HPRT* | F: TGACACTGGCAAAACAATGCA; R: GGTCCTTTTCACCAGCAAGCT

Values were normalised to *HPRT* expression using the $\Delta C_T$ method.

## Native and denaturing polyacrylamide gel electrophoresis (PAGE)

HEK293T, CHO and B3 cells were transiently transfected with pcDNA3.1-*CD274-L2A* in serum-free media. Supernatants from transfected and untransfected cells were centrifuged at 14,000 rpm for 30 min at 4°C and pellets were resuspended in sample loading buffer with or without SDS and 2-

mercaptoethanol (β-ME) and heat denaturation. Samples were run on a 4–20% polyacrylamide gel by native or SDS-PAGE, stained with Coomassie Brilliant Blue overnight and visualized on an Amersham Imager 600 (GE Healthcare).

## Statistical analyses

Statistical comparisons were made using GraphPad Prism 7 (GraphPad Software) or SigmaPlot 14.0. Parametric comparisons of normally distributed values that satisfied the variance criteria were made by unpaired Student's $t$-tests or One Way Analysis of variance (ANOVA) tests. Data that did not pass the variance test were compared with non-parametric two-tailed Mann–Whitney Rank Sum tests or ANOVA on Ranks tests. p values are indicated by asterisks as follows: *$0.01 < p < 0.05$; **$0.001 < p < 0.01$; ***$0.0001 < p < 0.001$; ****$p<0.0001$. Hierarchical clustering and heatmap production were performed with Qlucore Omics Explorer 3.3 (Qlucore).

## Acknowledgements

We are grateful for assistance from the Scientific Computing, Cell Services, Flow Cytometry, Biological Research, High Throughput Screening and Light Microscopy facilities at the Francis Crick Institute. The results shown here are in whole or part based upon data generated by the TCGA Research Network (http://cancergenome.nih.gov). The Genotype-Tissue Expression (GTEx) Project was supported by the Common Fund of the Office of the Director of the National Institutes of Health, and by NCI, NHGRI, NHLBI, NIDA, NIMH, and NINDS. This work benefited from data assembled by the CCLE consortium. This work was supported by the Francis Crick Institute (FC001099), which receives its core funding from Cancer Research UK, the UK Medical Research Council, and the Wellcome Trust; and by the Wellcome Trust (102898/B/13/Z).

## Additional information

### Funding

| Funder | Grant reference number | Author |
|---|---|---|
| Francis Crick Institute | 10099 | Kevin W Ng<br>Jan Attig<br>George R Young<br>Eleonora Ottina<br>George Kassiotis |
| Wellcome | 102898/B/13/Z | George Kassiotis |

The funders had no role in study design, data collection and interpretation, or the decision to submit the work for publication.

### Author contributions

Kevin W Ng, Data curation, Formal analysis, Investigation, Methodology, Writing—original draft; Jan Attig, George R Young, Data curation, Formal analysis, Investigation, Writing—original draft; Eleonora Ottina, Investigation; Spyros I Papamichos, Conceptualization, Formal analysis, Investigation; Ioannis Kotsianidis, Supervision, Project administration; George Kassiotis, Conceptualization, Supervision, Funding acquisition, Writing—original draft, Project administration

### Author ORCIDs

Kevin W Ng (iD) https://orcid.org/0000-0003-1635-6768
Jan Attig (iD) http://orcid.org/0000-0002-2159-2880
Spyros I Papamichos (iD) http://orcid.org/0000-0001-7119-0647
George Kassiotis (iD) https://orcid.org/0000-0002-8457-2633

### Ethics

Human subjects: This study was reviewed and approved by The Francis Crick Institute's Human Ethics Group and all experiments were carried out in accordance with the United Kingdom's Human

Tissue Act (2004). All participants provided written informed consent prior to participation in the study.

Animal experimentation: All animal experiments were approved by the ethical committee of the Francis Crick Institute and conducted according to local guidelines and UK Home Office regulations under the Animals Scientific Procedures Act 1986 (ASPA) (licence number: PCD77C6D0).

## Decision letter and Author response

Decision letter https://doi.org/10.7554/eLife.50256.SA1
Author response https://doi.org/10.7554/eLife.50256.SA2

## Additional files

### Supplementary files
• Transparent reporting form

### Data availability
All data generated or analysed during this study are included in the manuscript and supporting files. Source data files have been provided for Figure 1G, Figure 1—figure supplement 3, and Figure 3B.

The following previously published datasets were used:

| Author(s) | Year | Dataset title | Dataset URL | Database and Identifier |
|---|---|---|---|---|
| Linsley PC, Speake C, Whalen E, Chaussabel D | 2014 | Next generation sequencing of human immune cell subsets across diseases | https://www.ebi.ac.uk/ena/data/view/PRJNA258216 | European Nucleotide Archive, SRP045500 |

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
