## [Decision Letter]

**Acceptance summary:**

The PD-1 pathway is an important brake on the immune system. PD-L1 is a cell surface ligand that activates PD-1 on T cells, and both proteins are key targets in cancer immunotherapy. Kassiotis et al. report that the acquisition of a jumping gene in the PD-L1 gene created a shortened isoform of PD-L1. This new isoform is soluble and antagonize the function of full length PD-L1 in cells and in animals. This ancient genetic insertion is retained in many species but lost in others, providing an a compelling example of how the evolutionary battle between mammalian genomes and jumping genes diversifies and reshapes immunity.

**Decision letter after peer review:**

Thank you for submitting your article "Soluble PD-L1 generated by endogenous retroelement exaptation is a receptor antagonist" for consideration by *eLife*. Your article has been reviewed by three peer reviewers, including Howard Y Change as the Reviewing Editor and Reviewer #1, and the evaluation has been overseen by Satyajit Rath as the Senior Editor.

The reviewers have discussed the reviews with one another and the Reviewing Editor has drafted this decision to help you prepare a revised submission.

Summary:

The authors revealed that the exaptation of endogenous retroelement Line-2A in PD-L1 generates alternatively spliced variant as a soluble PD-L1 (sPD-L1). They further characterized the function of this sPD-L1 and identified a role of sPD-L1 in blocking the inhibitory activity of membrane-bound PD-L1.

Essential revisions:

1) The presence of a soluble PD-L1 isoform (sPD-L1) is known. The novelty of this work is the recognition that the presence of intronic Line-2A produces the soluble PD-L1 by omitting splicing site at the end of exon4 and providing the terminal exon and polyA. However, this conclusion lacks convincing evidence. To confirm the role of Line-2A in alternative splicing of PD-L1, it would be essential to delete intronic Line-2A in PD-L1 to test if the splicing variant of sPD-L1 is decreased. Hassounah et al., 2019) showed that this soluble PD-L1 is generated by the integration of HPV at the intron after exon 4 which is located in Line-2A region. Given that a loss of sPD-L1 variant in mice and the difference of Line-2A between mice and human, it would be better to map the key segment of Line-2A that controls the alternative splicing to distinguish if this effect is from HPV integration site or other sites in Line-2A. In addition, in Figure 1F, the authors showed gene tracks of different splicing variant. It would be better to design qPCR primer specifically targeting the terminal exon provided by Line-2A to confirm this sPD-L1 variant.

2) The authors claim that the L2A was exapted for a beneficial function. However, a much more rigorous evolutionary analysis is necessary to support this idea. For example, in Figure 2F, the authors conclude that the higher sequence similarity of the hominid L2A insertion to the consensus L2A reflects purifying selection on that element in hominids. But, this could also reflect a globally faster mutation rate in rodents, rather than purifying selection at this specific insertion. This evolutionary analysis must be substantially improved to support the idea this insertion was exapted, by including other L2A elements including ones expected to be neutrally evolving, and other nearby intronic sites. Furthermore, while the authors state that canonical splice/polyA sites are conserved in other species at the sequence level, it would be much more convincing if there were evidence that the sPD-L1 transcript is indeed produced in neutrophils in other species.

3) However, they include no in vivo studies to support these results. This is important to interpret the significance of sPD-L1. For example, an experiment examining administration of sPD-L1 in tumor bearing mice (or some other model) alone and in combination with anti-PD-1 mAb. Further, it would be interesting to see the effect of combination with anti-PD-L1 mAb – as the authors report, there has been a recent study showing sPD-L1 may interfere with the effect of anti-PD-L1 immunotherapy. These experiments would demonstrate the importance of sPD-L1 in the PD-L1 – PD-1 axis in immunity and facilitate therapeutic application. Overall, this paper offers some novel insight for basic biology and its potential importance in immunity and therapy. But unfortunately, they did not demonstrate its relevance for normal function or therapy in this paper.

4) In Figure 4-5, the authors characterized the function of sPD-L1 by overexpressing of this sPD-L1 variant. To test the role of endogenous sPD-L1 in T cell inhibition, it would be better to use a cell line that has highly expressed sPD-L1 (like Ramos or HBL-1 showed in Figure 3C) and delete the Line-2A as a control to compare the T cell inhibition. In addition, several studies showed that sPD-L1 has a T cell inhibitory activity, which is opposite to the author's findings. Although the authors suggested that the discrepancy is due to artificial Fc fusion in those studies, Mahoney et al., 2019) showed the T cell inhibitory activity of sPD-L1 without using Fc fusion. Thus it is essential to test the function of endogenous sPD-L1 to avoid the discrepancy and highlight the physiological function of sPD-

L1.

5) As the authors mentioned, many of the reported experiments for CD274-L2A have been performed previously in the cited Hassounah et al., 2019 and Mahoney et al., 2019, including the TCGA and CCLE transcriptome analysis, whether it inhibits T cell function, and whether it binds to PD-1. The authors find some contradictory results compared to published studies, and did a good job discussing the possible reasons for the inconsistency. However, the authors should perform more experiments to substantiate these differences, as these differences are crucial for the novelty of the study. Examples: a) The authors reported weak correlation between expressions of CD274-L2A and the full-length variant 1 in both healthy tissues and different TCGA cancers in Figure 3A, which is inconsistent with previous results. The authors mentioned that such difference is likely due to different RNA-seq quantitation methods, but they should clearly state what's the difference, and why their method is better. The correlation analysis should be performed at individual level within each tissue, rather than being performed using the averages of different tissue types, which is apparently depicted in Figure 3A.

b) Figure 4B showed monomeric sPD-L1 at ~28 kDa, and a tetramer (low quality western band, but if I believe what the authors claim rather than the reported dimer in the previous paper. Instead, Figure 1D of the cited Mahoney et al., 2019, and Figure 3C of the cited Hassounah et al., 2019, reported monomeric and dimeric forms at ~40 and ~80 kDa in the reducing, and non-reducing condition, respectively. The authors mentioned that the inconsistency could be due to different cell lines used in different studies, but they should perform experiments in the cell lines used in previous reports to show if this is the case.

---

## [Author Response]

Essential revisions:1) The presence of a soluble PD-L1 isoform (sPD-L1) is known. The novelty of this work is the recognition that the presence of intronic Line-2A produces the soluble PD-L1 by omitting splicing site at the end of exon4 and providing the terminal exon and polyA. However, this conclusion lacks convincing evidence. To confirm the role of Line-2A in alternative splicing of PD-L1, it would be essential to delete intronic Line-2A in PD-L1 to test if the splicing variant of sPD-L1 is decreased. Hassounah et al., 2019) showed that this soluble PD-L1 is generated by the integration of HPV at the intron after exon 4 which is located in Line-2A region. Given that a loss of sPD-L1 variant in mice and the difference of Line-2A between mice and human, it would be better to map the key segment of Line-2A that controls the alternative splicing to distinguish if this effect is from HPV integration site or other sites in Line-2A. In addition, in Figure 1F, the authors showed gene tracks of different splicing variant. It would be better to design qPCR primer specifically targeting the terminal exon provided by Line-2A to confirm this sPD-L1 variant.

We have now included experiments where we have specifically deleted this intronic L2A element as the reviewer suggests. To this end, we constructed a CD274 minigene, incorporating either the entire intron 4, which contains the exonised L2A, or an intron 4 with the L2A deleted. Introduction of the reference intron 4 to the CD274 cDNA restored production of sPD-L1 protein, whereas deletion specifically of the L2A element in this minigene abolished this production. These results establish the critical requirement for the L2A element in the production of sPD-L1 and are shown in the new Figure 1—figure supplement 2A-C. The requirement for the L2A element is also supported by new data on sPD-L1 production by cells from other species, according to the integrity of the L2A and splice and polyadenylation sites (new Figure 2—figure supplement 1C-D).

The reviewer might have misinterpreted the study by Hassounah et al. We should clarify that an integration of HPV16 was previously found in intron 4 of the CD274 gene in a single case of head and neck squamous cell carcinoma (Parfenov et al., PNAS 2014). This was a somatic, likely extrachromosomal event, where the entire CD274 gene, HPV16 insertion and part of the upstream PLGRKT gene underwent structural rearrangement, amplification, and overexpression.

It was this finding that inspired Hassounah et al. to look for novel variants transcribed from the CD274 locus, which allowed then to independently identify the CD274-L2A transcript. It is also interesting to note that our inspection of this somatic HPV16 integration shows that it happened right in the middle of the L2A element, replacing its function and highlighting its importance. We must reiterate, however, that the HPV16 integration is a somatic event in a single tumour and no HPV integration exists in the human genome, nor any effect of HPV can be expected. In contrast, the L2A integration is fixed in the human germline (and indeed in that of other mammals) and is essential for the production of the CD274-L2A transcript and of the sPD-L1 protein in unmutated cancers and healthy tissues.

We had indeed verified and quantified expression of the CD274-L2A transcript in human cells by qRT-PCR using primers specific to this transcript (with the reverse primer complementary to the L2A element). These results were shown in the original Figure 3C. We have now included similar qRT-PCR validation in non-human primate cells (Figure 2—figure supplement 1C-D).

2) The authors claim that the L2A was exapted for a beneficial function. However, a much more rigorous evolutionary analysis is necessary to support this idea. For example, in Figure 2F, the authors conclude that the higher sequence similarity of the hominid L2A insertion to the consensus L2A reflects purifying selection on that element in hominids. But, this could also reflect a globally faster mutation rate in rodents, rather than purifying selection at this specific insertion. This evolutionary analysis must be substantially improved to support the idea this insertion was exapted, by including other L2A elements including ones expected to be neutrally evolving, and other nearby intronic sites. Furthermore, while the authors state that canonical splice/polyA sites are conserved in other species at the sequence level, it would be much more convincing if there were evidence that the sPD-L1 transcript is indeed produced in neutrophils in other species.

We agree with the reviewer and appreciate the suggestions. Higher apparent conservation of the L2A in humans than in mice does not prove purifying selection in the former and it is, as the reviewer points out, more likely caused by faster evolution of mice. It is nevertheless interesting to note that the mutations in rodents affect specifically the production of CD274-L2A.

Overall, the L2A fragment in the CD274 seems to be evolving at an average rate in humans (divergence: 30.3% and 27.3%; deletions: 7.3% and 7.8%; insertions: 0.0% and 5.0% for the CD274-L2A and the average of all other L2A elements of similar length in the genome, respectively). However, as indicated by original alignment in Figure 2A-B, this part of intron 4 appears as well conserved in hominids as the coding exons. We have now extended this analysis over the entire CD274 locus (shown in new Figure 2G), to show that in 10 primate species, the 100 nucleotides covering the exonised part of intron 4 and embedded L2A element were as conserved as the coding exons, contrasting with the rest of *CD274* intron 4, other *CD274* introns and non-coding exons.

Moreover, following the reviewer’s suggestion, we have now examined whether or not the CD274-L2A transcript is produced as predicted by the splice and polyadenylation sequences in other species. As primary neutrophils (some of the shortest-lived cells of the body) from other species are not available, we used green monkey cells lines (CV-1 and Vero), rabbit cell line (R9ab) and mouse cell lines (EL4 and MCA-38), all of which express the respective full-length CD274. However, whereas CD274-L2A was readily detectable in non-human primate cells, it was absent from rabbit or mouse cells, as predicted by splice site and L2A sequences.

*3) However, they include no* in vivo *studies to support these results. This is important to interpret the significance of sPD-L1. For example, an experiment examining administration of sPD-L1 in tumor bearing mice (or some other model) alone and in combination with anti-PD-1 mAb. Further, it would be interesting to see the effect of combination with anti-PD-L1 mAb – as the authors report, there has been a recent study showing sPD-L1 may interfere with the effect of anti-PD-L1 immunotherapy. These experiments would demonstrate the importance of sPD-L1 in the PD-L1 – PD-1 axis in immunity and facilitate therapeutic application. Overall, this paper offers some novel insight for basic biology and its potential importance in immunity and therapy. But unfortunately, they did not demonstrate its relevance for normal function or therapy in this paper.*

As the reviewer suggested, we have now investigated the role of sPD-L1 in an in vivo tumour setting using the MCA-38 colon adenocarcinoma cell line, which has been described to be sensitive to PD-1/PD-L1 blockade. MCA-38 cells expressing human or murine sPD-L1 phenocopied the delayed tumour growth with exogenous PD-L1 blockade when subcutaneously transplanted into immunocompetent mice despite having equivalent in vitro growth rates. These results are displayed in Figures 6A-B and Figure 6—figure supplement 1. Whilst these experiments support in vivo biological activity for sPD-L1 in tumour bearing mice, consistent with our in vitro data, we do acknowledge that this is one of many settings in which the antagonistic activity of sPD-L1 could be important. To fully elucidate the physiological function of sPL1 beyond the tumour setting, we would have to study a mouse model and, as mice naturally lack production of sPD-L1, we would first have to restore it in by re-introducing the ancestral intron 4 into the mouse germline. This would allow the study of the physiological function of sPD-L1 (at physiological levels and regulation) in processes, such as immune development, autoimmunity, viral infection or placentation, where preliminary results suggest it might play an important role. However, these experiments are far beyond the scope of the current manuscript.

4) In Figure 4-5, the authors characterized the function of sPD-L1 by overexpressing of this sPD-L1 variant. To test the role of endogenous sPD-L1 in T cell inhibition, it would be better to use a cell line that has highly expressed sPD-L1 (like Ramos or HBL-1 showed in Figure 3C) and delete the Line-2A as a control to compare the T cell inhibition. In addition, several studies showed that sPD-L1 has a T cell inhibitory activity, which is opposite to the author's findings. Although the authors suggested that the discrepancy is due to artificial Fc fusion in those studies, Mahoney et al., 2019) showed the T cell inhibitory activity of sPD-L1 without using Fc fusion. Thus it is essential to test the function of endogenous sPD-L1 to avoid the discrepancy and highlight the physiological function of sPD-L1.

Although we produce sPD-L1 by expression in cell lines that do not normally express it, we should note that we have used it at more physiological levels and ratio with the full-length form than the previous studies. For their immunosuppression assays, Mahoney et al. used a recombinant purified form of a HIS-tagged, rather than an Fc-fusion of sPD-L1, and reported only weak suppression at concentrations between 10 and 20 micrograms/ml. We have used native sPD-L1 at 2 nanograms/ml (i.e. up to 10,000 times lower amounts) or lower and in par with the physiological concentration of sPD-L1. At these levels, we have consistently failed to detect suppressive activity, whereas antagonistic activity was still readily measurable. We should also note that the cell lines with high expression of sPD-L1 do not necessarily reflect physiology. For example, B cell leukaemias and derived cell lines (like Ramos or HBL-1), very frequently exhibit extensive structural rearrangement and amplification of the CD274 locus (e.g. Wessendorf et al., 2007; Green et al., 2010; Rosenwald et al., 2003). This not only affects the physiological ratio of membrane-bound, exosomal and soluble PD-L1 (as part of the transformation process), but also creates the practical problem of correct targeting of the various rearranged and amplified CD274 copies. Consequently, our effort to target the endogenous CD274 locus in such cell lines have been thwarted by its rearrangements and amplification. Complete elucidation of the physiological function of sPD-L1 will therefore necessitate the creation of a mouse strain, where the ancestral allele of Cd274 is restored. As we outlined in our reply to comment 3 above, this would be the only way to study of the physiological function of sPD-L1 (at physiological levels and regulation) in processes, such as immune development, autoimmunity, viral infection or placentation.

5) As the authors mentioned, many of the reported experiments for CD274-L2A have been performed previously in the cited Hassounah et al., 2019 and Mahoney et al., 2019, including the TCGA and CCLE transcriptome analysis, whether it inhibits T cell function, and whether it binds to PD-1. The authors find some contradictory results compared to published studies, and did a good job discussing the possible reasons for the inconsistency. However, the authors should perform more experiments to substantiate these differences, as these differences are crucial for the novelty of the study. Examples: a) The authors reported weak correlation between expressions of CD274-L2A and the full-length variant 1 in both healthy tissues and different TCGA cancers in Figure 3A, which is inconsistent with previous results. The authors mentioned that such difference is likely due to different RNA-seq quantitation methods, but they should clearly state what's the difference, and why their method is better. The correlation analysis should be performed at individual level within each tissue, rather than being performed using the averages of different tissue types, which is apparently depicted in Figure 3A.b) Figure 4B showed monomeric sPD-L1 at ~28 kDa, and a tetramer (low quality western band, but if I believe what the authors claim rather than the reported dimer in the previous paper. Instead, Figure 1D of the cited Mahoney et al., 2019, and Figure 3C of the cited Hassounah et al., 2019, reported monomeric and dimeric forms at ~40 and ~80 kDa in the reducing, and non-reducing condition, respectively. The authors mentioned that the inconsistency could be due to different cell lines used in different studies, but they should perform experiments in the cell lines used in previous reports to show if this is the case.

Overall our results are in agreement with those reported by Hassounah et al., 2019 and Mahoney et al., 2019. For example, we all find that CD274-L2A encodes a soluble, secreted PD-L1 protein, this protein is multimeric (tetrameric in our case) and retains receptor binding, its expression is detected in healthy tissues and it is elevated in several cancer types. The single most important difference is that we find that CD274-L2A-encoded sPD-L1 is a receptor antagonist, a possibility that was not examined in the previous studies.

There are, however, other methodological differences that the reviewer perceptively identified:

a) Indeed, our analysis of RNA-seq data from a multiple cancer types and healthy tissues indicates only weak correlation between the expression of full-length and truncated PD-L1 variants. We do observe a correlation in the same direction as the previous two studies, simply not as strong and certainly not as statistically significant. There are several differences, which are now discussed in the text. In their analysis, Mahoney et al. used a non-standard method for quantitation of the two isoforms based of the number of reads mapping uniquely to the parts of the two isoforms that are different, and normalised by the read count in the RNA-seq library (a variant of RPKM). Similarly, Hassounah et al. used the ratio of exon 4 (shared exon) to exon 5 (not present in CD274-L2A) as a proxy for CD274-L2A expression, which they calculated again using the RPKM method. Instead, we used the TPM method, originally proposed as an alternative to RPKM due to inaccuracy in RPKM measurements (Wagner et al., 2012). Also, using our extended assembly, we took into account any other exon-sharing transcript produced by the *CD274* locus and normalised expression by the total transcript count (integral part of TPM, but not RPKM calculations). We have now added the new Figure 3—figure supplement 1, depicting the correlation performed at individual level within each tissue, as per reviewer’s suggestion. We focused on tissues/cancers with sufficient expression and number of samples to enhance our ability to detect even weak correlations. The new analysis corroborates our earlier analysis, in that there seems to be little correlation between expression of the two variants. For example, healthy lung expresses primarily the full-length isoform, whereas in LUAD the balance shifts in favour of the truncated isoform.

b) We agree with the reviewer that our original Western blots were low quality. This is due to the poor ability of the available antibodies to detect the native form of PD-L1. We have tested all widely-used PD-L1 clones suitable for immunoblot (E1L3N, Cell Signaling; D8T4X, Cell Signaling; 28-8, Abcam), and none showed cleaner bands under native conditions. Of note, Hassounah et al. use clone 28-8 (Abcam) only under denaturing conditions and did not show any data under native conditions. Mahoney et al. showed data under native conditions, but used an in-house clone 368A.5A4 that is not commercially available and could not be tested here. As far as we can tell, this is the only clone reported to detect PD-L1 under native conditions.

To conclusively determine the molecular form of native sPD-L1, without the need for an antibody compatible with native Western blot, we have now expressed native sPD-L1 (without any tags or Fc fusions), in three separate cell lines (including those used in the previous studies), which we grew in protein-free media. This allowed us to directly and clearly visualise native, as well as denatured sPD-L1 by Coomassie stain of native and denaturing PAGE gels. These results are now shown in the modified Figure 4B and support our previous conclusion based on Western blotting, revealing that the native form of sPD-L1 has a molecular weight consistent with a tetramer (~120 kDa) and the monomeric form a molecular weight of ~30 kDa, close to its theoretical weight of 28 kDa.